# Microglia modulate stable wakefulness via the thalamic reticular nucleus in mice

Hanxiao Liu[1], Xinxing Wang[1], Lu Chen[1], Liang Chen[1], Stella E. Tsirka[2], Shaoyu Ge[1] & Qiaojie Xiong[1 ✉]

Microglia are important for brain homeostasis and immunity, but their role in regulating vigilance remains unclear. We employed genetic, physiological, and metabolomic methods to examine microglial involvement in the regulation of wakefulness and sleep. Microglial depletion decreased stable nighttime wakefulness in mice by increasing transitions between wakefulness and non-rapid eye movement (NREM) sleep. Metabolomic analysis revealed that the sleep-wake behavior closely correlated with diurnal variation of the brain ceramide, which disappeared in microglia-depleted mice. Ceramide preferentially influenced microglia in the thalamic reticular nucleus (TRN), and local depletion of TRN microglia produced similar impaired wakefulness. Chemogenetic manipulations of anterior TRN neurons showed that they regulated transitions between wakefulness and NREM sleep. Their firing capacity was suppressed by both microglial depletion and added ceramide. In microglia-depleted mice, activating anterior TRN neurons or inhibiting ceramide production both restored stable wakefulness. These findings demonstrate that microglia can modulate stable wakefulness through anterior TRN neurons via ceramide signaling.

[1] Department of Neurobiology and Behavior, Stony Brook University, Stony Brook, NY, USA. [2] Department of Pharmacological Sciences, Stony Brook University, Stony Brook, NY, USA. ✉email: qiaojie.xiong@stonybrook.edu

Microglia account for 10–15% of all cells in the adult brain. They not only serve as the main defense against external/internal immune challenges in the central nervous system, but also participate in synaptic formation and elimination, neurogenesis, and learning and memory[1–3]. Accumulating evidences indicate a correlation between microglia activity and sleep/wake states: microglial morphology and activity exhibit distinct dynamics based on the time of day and brain states[4–7], and sleep dysfunction is prevalent in diseases related to abnormal microglial activity, such as Alzheimer's disease and parasitic infections[8,9]. Although it has been shown that sleep deprivation altered microglia activation[10–12], it remains unknown, in the opposite direction, whether and how microglia regulate sleep/wake.

Wakefulness and sleep are essential components of daily life. Sleep can be broadly categorized into two states on the basis of electroencephalogram (EEG) and electromyogram (EMG) recordings: rapid eye movement (REM) sleep and nonrapid eye movement (NREM) sleep[13,14]. Many neural circuits have been implicated in the regulation of sleep/wake states, most of which are in subcortical brain regions[15,16]. The thalamic reticular nucleus (TRN), a group of GABAergic interneurons (98% of which are parvalbumin positive) that provide inhibitory input to the thalamus, is critical for regulating sleep rhythm and attention[17–19]. The TRN has been anatomically divided into two main sections based on projection targets: the anterior TRN (aTRN), involved primarily in limbic function, and the posterior TRN which is involved with processing sensory information[20–24]. Several recent studies have shown that aTRN neurons display differential firing rates during wake, NREM sleep, and REM sleep[17,20,25,26]. However, the cellular and molecular mechanisms underlying these differential firing rates remain unknown.

To determine whether and how microglia modulate sleep, we used a transgenic mouse line in which the diphtheria toxin (DT) receptor is conditionally expressed in microglia for the purpose of microglial depletion[27,28]. We found that microglial depletion dramatically decreased stable wakefulness due to an increase in transitions between wakefulness and NREM sleep. Through metabolic screening and immunostaining, we revealed that the diurnal dynamics of brain ceramide was highly correlated with sleep-wake behavior, and this correlation was abolished in microglia-depleted mice. Microglia in the TRN are sensitive to changes of ceramide levels, and microglial depletion or addition of ceramide decreased aTRN neuronal activity. Importantly, the chemogenetic suppression of aTRN neuronal activity promoted increased transitions between wakefulness and NREM sleep in control mice, whereas chemogenetic activation of aTRN neuronal activity in microglia-depleted mice rescued impaired stable wakefulness. Together, these findings indicate that microglia can modulate stable wakefulness through anterior TRN neurons via ceramide signaling.

## Results

**Microglial depletion reduced stable wakefulness by increasing transitions between wakefulness and NREM sleep.** In contrast to their immune-surveillance function, the regulatory role of microglia in vigilance remains poorly understood. In this cohort of experiments, we examined the role of microglia in regulating wakefulness and sleep.

Changes in wakefulness and sleep were assessed in adult mice with or without microglia depletion. To genetically deplete microglia, as illustrated in Fig. 1a or as previously reported[27,28], we administered two doses of tamoxifen (TAM, 0.5 mg/g) over 48 h for initiation of cre-dependent *iDTR* transgenes recombination under *CX3CR1* promoter in a group of adult

CX3CR1$^{Cre-ERT2/+}$:R26$^{iDTR/+}$ mice. Four weeks after this induction (defined as day 0), we injected DT (i.p., once per day for three consecutive days) to deplete DT receptor-expressing cells. Another group of mice with the same genetic background and tamoxifen treatment received 3 days of saline injection instead of DT to serve as controls. Microglia in the brain have much longer lifespan (>8 weeks) than other CX3CR1-expressing cells in periphery (<2 weeks)[29,30], which allows a time window to specifically deplete microglia in the brain but not affect other CX3CR1-expressing cells in the periphery. Peripheral CX3CR1-expressing cells with DT receptor expression disappeared at 4 weeks after tamoxifen treatment due to their quick renewal[27,28,30,31]. Consistently, we found at 4 weeks after tamoxifen administration, DT injection led to a prominent decrease in density of CX3CR1-expressing cells in the brain (i.e., microglia), but not in kidney and spleen (Fig. 1b). To further validate depletion efficiency in microglia, DT-injected mice were sacrificed on day 0 (prior to DT injection), day 4, or day 6. We performed immunostaining with one antibody against GFP (EYFP expression was driven by *CX3CR1* promoter in this mouse line[4]), and another against Iba1, a microglia/macrophage-specific calcium-binding protein[32]. We then counted the number of Iba1+/GFP+ microglia in different brain regions. We found that on day 4 (one day after the last DT injection), the numbers of microglia decreased sharply to low levels in various brain regions compared to those on day 0 (Fig. 1c). Note that microglia depletion had no effect on body weight, locomotion and other cell populations in the brain, such as astrocytes and neurons (Supplementary Fig. 1a, b and d–f), and saline injections had no detectable effect on microglial density (Supplementary Fig. 1c).

Next, we analyzed vigilance states in microglia-depleted mice. Both EEG and EMG recordings were performed before and after DT injections for 6 h during the day (Zeitgeber time 3–9) and 6 h during the night (Zeitgeber time 15–21) at 4 weeks after tamoxifen administration (Fig. 1a). Wakefulness, NREM sleep, and REM sleep epochs were measured based on EEG/EMG recordings as previously described[13,15]. We then compared sleep architecture between day 0 (baseline) and day 4, for both day and night, in the saline or DT-injected CX3CR1$^{Cre-ERT2/+}$:R26$^{iDTR/+}$ mice (MG$^{DTR+}$ Veh. and MG$^{DTR+}$ DT respectively; microglia were depleted in MG$^{DTR+}$ DT group) and DT-injected CX3CR1$^{Cre-ERT2/+}$:R26$^{iDTR/-}$ mice (MG$^{DTR-}$ DT) which lack of *iDTR* transgene (Fig. 1a, d). We first validated that neither TAM nor DT treatment per se had any detectable effect on sleep architecture (Supplementary Fig. 2a, b). We next found that microglia-depleted mice exhibited decreased total wakefulness durations (82.1 ± 5.5% of day 0) and increased total NREM sleep durations (133.7 ± 8.1% of day 0) on the night of day 4 (Fig. 1d, Supplementary Fig. 2d, 3a–c). Note that there was no change between day 0 and day 4 in saline-injected MG$^{DTR+}$ mice and DT-injected MG$^{DTR-}$ mice (Fig. 1d, Supplementary Fig. 2b–c, 3a–c), indicating that the change of sleep architecture is due to microglia depletion but not DT or saline injection per se. Further analysis revealed a sharp decrease in wakefulness-bout durations at night (Fig. 1e), mainly in stable wakefulness but not quick arousal (Fig. 1f). To further understand these vigilance dynamics, we analyzed the transitions between different brain states. The numbers of transitions from wake to NREM sleep (WN) and from NREM sleep to wake (NW) were significantly increased in microglia-depleted mice (Fig. 1g, Supplementary Fig. 3b). Interestingly, microglial depletion had no effect on REM sleep durations (Fig. 1d–e, Supplementary Fig. 2d, 3a–c) and transitions from NREM sleep to REM sleep (NR) and REMs-to-wake (RW) (Fig. 1g). The vigilance dynamics between three groups of mice were comparable prior to microglial depletion (Day 0)

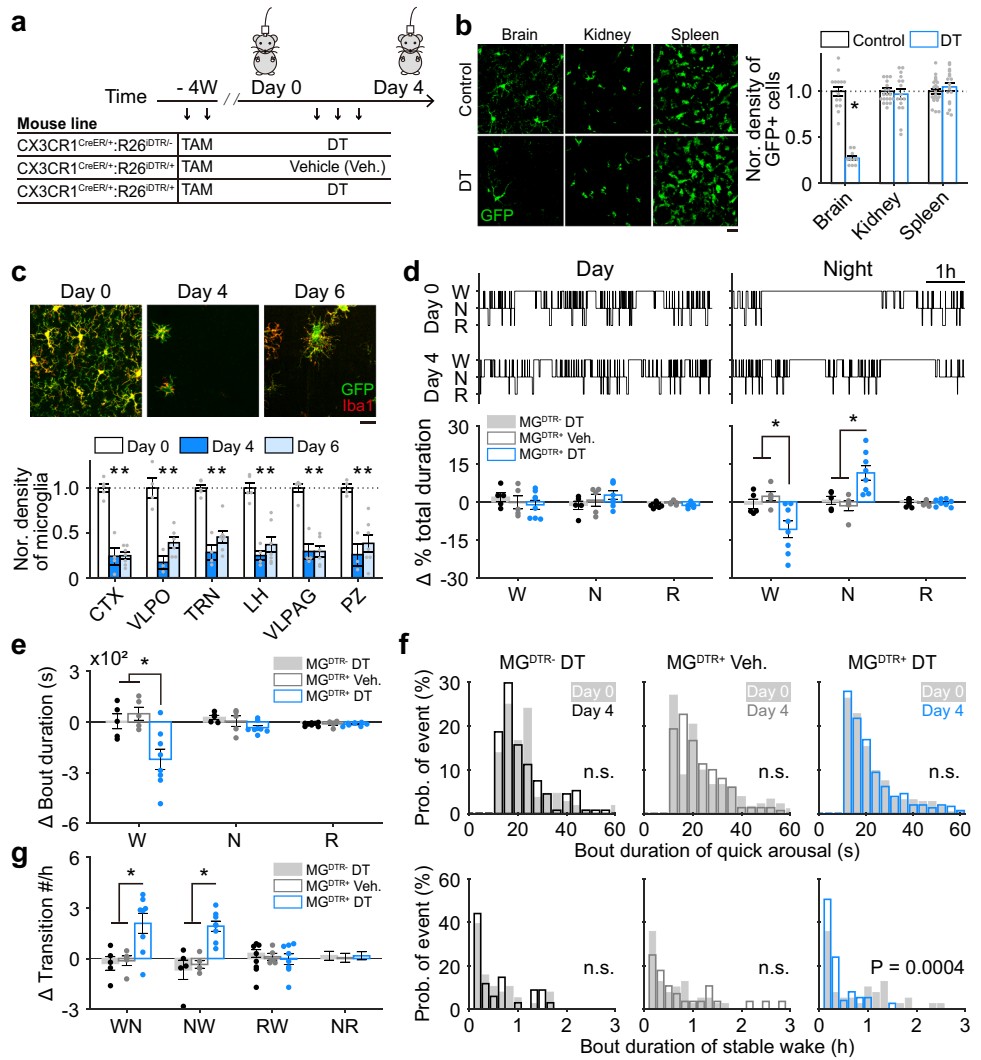

**Fig. 1 Microglia are essential for stable wakefulness at night. a** Microglia were noninvasively depleted by injecting DT (i.p.) into CX3CR1^Cre-ERT2/+: R26^iDTR/+ transgenic mice. EEG/EMG signals were recorded before and after injection for 6 h during the middle of day and night periods. Three groups of mice were included: CX3CR1^Cre-ERT2/+:R26^iDTR/− mice with tamoxifen (TAM) and DT injection (MG^DTR− DT), CX3CR1^Cre-ERT2/+:R26^iDTR/+ mice with TAM and vehicle injection (MG^DTR+ Veh.), CX3CR1^Cre-ERT2/+:R26^iDTR/+ mice with TAM and DT injection (MG^DTR+ DT). **b** Four weeks after TAM treatment, DT injection sharply decreased CX3CR1-expressing cells in the brain ($p = 1.1 \times 10^{-10}$), with no effect on kidney and spleen of CX3CR1^Cre-ERT2/+: R26^iDTR/+ mice. Control, 3 mice; $n = 14$ images from brain, $n = 20$ images from kidney, $n = 22$ images from spleen; DT, 2 mice; $n = 10$ images from brain, $n = 16$ images from kidney, $n = 18$ images from spleen, Each dot represents cell density obtained from one image. Scale bar, 20 μm. **c** DT depleted ~80% of microglia (cells co-labeled with GFP and Iba1) across the whole brain (day 0 vs day 4, $p < 0.01$ for all examined regions; day 0 vs day 6, $p < 0.01$ for all examined regions). Representative images show cortical microglia. Microglial density was normalized to day 0 values. CTX cortex, VLPO ventrolateral preoptic nucleus, TRN thalamic reticular nucleus, LH lateral hypothalamus, VLPAG ventrolateral periaqueductal gray, PZ parafacial zone. At least four images were collected for each brain region of one mouse, each dot represents cell density obtained from one mouse. day 0, $n = 5$ mice; day 4, $n = 4$ mice; day 6, $n = 7$ mice. Scale bar, 50 μm. **d, e** A hypnogram measured from one mouse during the day (top, left) and at night (top, right) before and after microglial depletion (W wakefulness, N NREM sleep, R REM sleep). Sleep architecture (bottom in **d**: total duration for each state; **e**: bout duration for each state) was compared before (day 0) and after (day 4) DT or vehicle injection for each animal, and the difference (day 4–day 0) was further compared between three groups of mice. Each dot represents the difference between day 0 and day 4 in one mouse. For Δ % total duration at night in **d**; W: $p = 0.025$ for MG^DTR− DT vs MG^DTR+ DT; $p = 0.005$ for MG^DTR+ Veh. vs MG^DTR+ DT; N: $p = 0.008$ for MG^DTR− DT vs MG^DTR+ DT; $p = 0.002$ for MG^DTR+ Veh. vs MG^DTR+ DT. For Δ bout duration in **e**; W: $p = 0.01$ for MG^DTR− DT vs MG^DTR+ DT; $p = 0.003$ for MG^DTR+ Veh. vs MG^DTR+ DT. MG^DTR+ DT, $n = 8$ mice; MG^DTR+ Veh., $n = 5$ mice. MG^DTR− DT, $n = 5$ mice. **f** Microglial depletion shortened wakefulness stability (bottom) but not quick arousals (top) at night. **g** Microglial depletion enhanced the number of wakefulness to NREM sleep (WN) and NREM sleep-to-wakefulness (NW) transitions at night, the transition from NREM sleep to REM sleep (NR) and REM sleep-to-wake were still intact (RW). WN: $p = 0.005$ for MG^DTR− DT vs MG^DTR+ DT; $p = 0.008$ for MG^DTR+ Veh. vs MG^DTR+ DT; NW: $p = 1.9 \times 10^{-4}$ for MG^DTR− DT vs MG^DTR+ DT, $p = 6.2 \times 10^{-4}$ for MG^DTR+ Veh. vs MG^DTR+ DT. MG^DTR+ DT, $n = 8$ mice; MG^DTR+ Veh., $n = 5$ mice. MG^DTR− DT, $n = 5$ mice. *$p < 0.05$; two-sided unpaired $t$-test for **b, c**; one-way ANOVA with Fisher's post-hoc test for **d**, **e**, and **g**; two-sided Kolmogorov–Smirnov test for **f**. Data are reported as mean ± SEM. See also Supplementary Figs. 1–3.

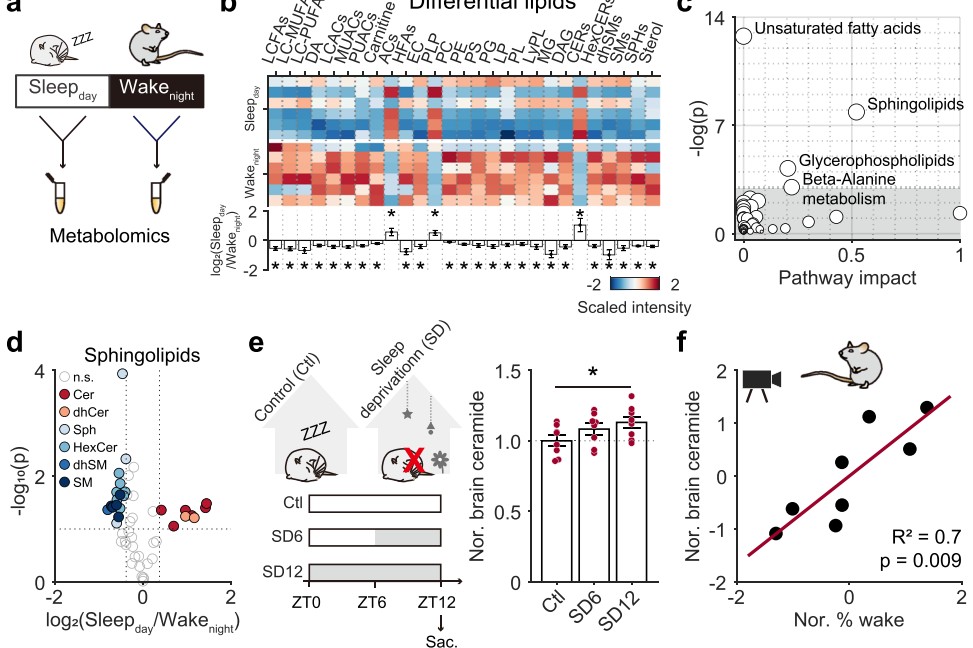

**Fig. 2 Circadian variation of ceramide correlated with sleep-wake behavior. a** Subcortical brain regions were collected during the day when mice were asleep (Sleep$_{day}$, $n = 6$ mice) and at night when mice were awake (Wake$_{night}$, $n = 6$ mice) for metabolomic analysis. **b** A heatmap of lipid species showing significant Sleep$_{day}$/Wake$_{night}$ expression differences. The $p$ value for each lipid was obtained by compare the difference between Sleep$_{day}$ and Wake$_{night}$. LCFAs, long chain saturated fatty acids, $p = 0.019$; LC-MUFAs long chain monounsaturated fatty acids, $p = 0.006$; LC-PUFAs long chain polyunsaturated fatty acids, $p = 0.016$; DA dicarboxylic fatty acids, $p = 0.01$; LCACs long chain saturated acyl carnitines, $p = 0.005$; MUACs monounsaturated acyl carnitines, $p = 0.004$; PUACs polyunsaturated acyl carnitines, $p = 0.022$; Carnitine, $p = 0.036$; ACs acylcholines, $p = 0.048$; HFAs dihydroxy fatty acids, $p = 0.008$; EC endocannabinoids, $p = 0.04$; PLP phospholipids, $p = 0.018$; PC phosphatidylcholines, $p = 0.01$; PE phosphatidylethanolamine, $p = 0.019$; PS phosphatidylserine, $p = 0.024$; PG phosphatidylglycerol, $p = 0.019$; LP lysophospholipid, $p = 0.026$; PL plasmalogen, $p = 0.034$; LyPL lysoplasmalogen, $p = 0.044$; MG monoacylglycerol, $p = 0.021$; DAG diacylglycerol, $p = 0.031$; CERs ceramides, $p = 0.04$; HexCERs hexosylceramides, $p = 0.018$; dhSMs dihydrosphingomyelins, $p = 0.043$; SMs sphingomyelins, $p = 0.021$; SPHs sphingosines, $p = 0.003$; Sterol, $p = 0.042$. **c** A pathway analysis indicated that lipid-related pathways were most affected by Sleep$_{day}$/Wake$_{night}$. **d** A volcano plot of measured sphingolipids; differentially represented metabolites with >1.3-fold change (Sleep$_{day}$ vs Wake$_{night}$) and p values <0.10 are indicated in color; n.s. nonsignificant. **e** Sleep deprivation promoted subcortical ceramide accumulation. Subcortical region was sampled at ZT12 (timepoint of light off). $p = 0.035$ for Ctl vs SD12. Ctl control mice, $n = 8$; SD6, mice with 6 h of sleep deprivation, $n = 8$; SD12, mice with 12 h of sleep deprivation, $n = 8$. **f** Tight correlation between brain ceramide level and wakefulness. Wakefulness was defined with video recording for 1 h before sampling (ZT12). Pearson's correlation, $p$ value was calculated using two-sided hypothesis test. Eight mice. *$p < 0.05$; two-sided unpaired $t$-test for **b, d**, and **e**. Data are reported as mean ± SEM. See also Supplementary Fig. 4.

(Supplementary Fig. 3d), indicating expression of DT receptor per se had no effect on sleep architecture.

These analyses revealed that microglial depletion induced a reduction of stable wakefulness at night-time by increasing wake to NREM sleep transitions, suggesting a potential role of microglia in modulating wakefulness and sleep behaviors.

**Microglial depletion abolished the diurnal variation of ceramide signaling that modulated stable wakefulness.** To explore how microglia potentially modulate wakefulness stability at night, we performed a metabolomics assay to analyze systemic factors. As mice generally sleep during daytime and are awake at night, we examined metabolite differences between brains collected from sleeping mice (Zeitgeber time 5–7) and those of awake mice (Zeitgeber time 17–19). Briefly, we collected brain tissues from subcortical regions that have been implicated in sleep and wakefulness regulation[13,15,16], and compared metabolite concentrations between brain tissues collected from sleeping mice during the day (Sleep$_{day}$) and awake mice at night (Wake$_{night}$) (Fig. 2a). A total of 155 metabolites exhibited significant concentration differences between Sleep$_{day}$ and Wake$_{night}$ (Supplementary Fig. 4a–c, Supplementary Table 1). Among these metabolites, the majority were lipid family members, including unsaturated fatty acids, sphingolipids, and glycerophospholipids

(Fig. 2b, c). Among the lipids, we found that ceramide, a bioactive metabolite, displayed a substantially lower concentration during the wakeful period (night-time) compared to the sleep period (daytime) (Fig. 2d, Supplementary Fig. 4d). This diurnal change of ceramide concentrations suggests a potential modulatory effect between ceramide and sleep: either sleep causes accumulation of ceramide, or high ceramide concentration promotes sleep. If it is the former case, sleep deprivation will block brain ceramide accumulation. We thus measured brain ceramide concentration following 6 h and 12 h of sleep deprivation (Fig. 2e). Interestingly, we observed a significant increase of ceramide following 12 h of sleep deprivation (Fig. 2e), suggesting that ceramide increases with stronger sleep drive. We further determined the relationship between arousal duration and ceramide concentration in naïve mice, and found that longer wakefulness before sampling was highly correlated with higher ceramide level (Fig. 2f). Taken together, these results indicated that ceramide positively correlates with sleep drive and may affect wakefulness stability via promoting sleep.

We then determined whether ceramide is involved in microglial modulation of wakefulness stability. First, we assessed whether microglia depletion affected ceramide concentration. We collected brain tissues from subcortical regions of microglia depleted and control mice with sampling standard consistent with

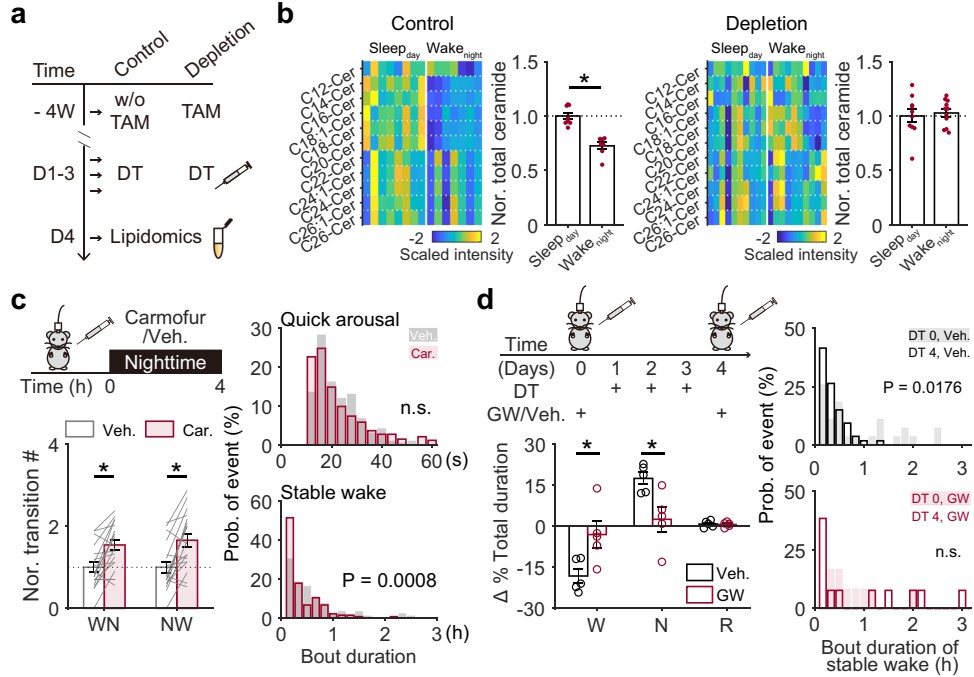

**Fig. 3 Ceramide-mediated microglial modulation of stable wakefulness. a** Experimental procedure for measurement of subcortical ceramide in CX3CR1$^{Cre-ERT2/+}$:R26$^{iDTR/+}$ transgenic mice. **b** Sleep$_{day}$/Wake$_{night}$ ceramide levels in microglia depleted (right; Sleep$_{day}$, $n = 10$ mice, Wake$_{night}$, $n = 10$ mice) and control mice (left; daytime, $n = 8$ mice; night-time, $n = 7$ mice). $p = 3.6 \times 10^{-5}$ for Sleep$_{day}$ vs Wake$_{night}$ in control mice. **c** An increase in nocturnal ceramide levels induced by carmofur administration facilitated state transitions between wakefulness and NREM sleep and decreased stable wakefulness in wild-type mice. Car. carmofur, Veh. vehicle. EEG/EMG signals were recorded for 4 h after administration at night ($n = 18$ pairs of recordings from 6 mice). Each line represents change in transition number between recording sessions with vehicle/carmofur injection collected from one mouse. Veh. vs Car.: $p = 5.1 \times 10^{-4}$ for WN; $p = 3.7 \times 10^{-4}$ for NW. **d** A decrease in ceramide by GW-4869 administration rescued stable wakefulness at night in microglia-depleted mice. GW-4869 or vehicle was injected three times before microglial depletion. Night-time EEG/EMG signals were recorded for 6 h after administration on day 0 (before depletion) and on day 4 (after depletion). Each dot represents the difference in % total duration for each brain state between day 0 and day 4 in one mouse. Veh. vs GW: $p = 0.032$ for W; $p = 0.026$ for N. GW, GW-4869 ($n = 5$ mice); Veh., vehicle ($n = 5$ mice). *$p < 0.05$; two-sided unpaired $t$-test for **b**, and left panel in **d**, two-sided paired $t$-test for left panel in **c**, two-sided Kolmogorov–Smirnov test for the right panels in **c** and **d**. Data are reported as mean ± SEM. See also Supplementary Fig. 4.

Fig. 2a and then performed smaller-scale lipidomic assays to measure ceramide concentrations (Fig. 3a). Consistent with the metabolomic study, we found that the total ceramide concentration in control mice was lower in Wake$_{night}$ than Sleep$_{day}$ (Fig. 3b). However, in microglia-depleted mice, this difference was abolished (Fig. 3b).

This led us to investigate whether manipulation of the night-time ceramide concentration would mimic, or mitigate, the microglia depletion effect on wakefulness stability in control mice and microglia-depleted mice, respectively. Indeed, when we increased ceramide in control mice at night via an i.p. injection of carmofur, an acid ceramidase inhibitor that can cross the blood-brain barrier and elevate brain ceramide levels 2 h after i.p. administration[33], we found a decrease in stable wakefulness (Fig. 3c, Supplementary Fig. 4e), which is similar to what we observed in microglia-depleted mice (Fig. 1f, g). In contrast, when ceramide was decreased in microglia-depleted mice by administering GW-4869, a neutral sphingomyelinase-2 inhibitor known to effectively block production of brain ceramide[34–37], we observed a restoration of wakefulness stability (Fig. 3d).

Taken together, we demonstrated that microglia may serve as important modulators of diurnal ceramide fluctuations in the brain to influence stable wakefulness at night.

**TRN microglia exhibited preferential sensitivity to brain ceramide levels, and local depletion of TRN microglia decreased stable wakefulness at night.** We next examined the possible neural circuit involved in the microglia-ceramide modulation of night-time wakefulness stability. Microglial morphology is tightly linked to their activity[6,7,38]. Activated microglia display shorter but thicker processes, larger cell bodies, and higher Iba1 expression[1,32,39]. Via morphological analysis of microglia in brain regions involved in sleep/wakefulness regulation[13,15,16] with the sampling standard described in Fig. 2a, we identified multiple subcortical regions showing diurnal morphological changes between Wake$_{night}$ and Sleep$_{day}$ mice, such as median preoptic area (MnPO), lateral preoptic area (LPO), thalamic reticular nucleus (TRN), suprachiasmatic nucleus (SCN) and dorsomedial hypothalamic nucleus (DM) (Fig. 4a). To explore whether the diurnal morphological difference of microglia in these brain regions is affected by changes in ceramide concentration, we elevated brain ceramide levels by directly applying exogenous C2-ceramide into mouse brains with i.c.v. injections and assessed microglial morphology in these brain regions with Iba1 staining at 3 h after administration (Fig. 4b). We first found that the Iba1 expression was specifically higher in TRN, a brain region that surrounding the thalamus and implicated in regulating sleep rhythm[24,40–42] (Fig. 4b, upper panel). We next revealed that TRN microglia retracted their processes in response to ceramide application (3 h after injection), whereas other examined regions showed no significant morphological change (Fig. 4b, lower panel). We further validated this result in alkaline ceramidase-3 knockout (Acer3$^{-/-}$) mice, that have higher brain ceramide levels than wild-type mice[43]. Consistently, microglia in Acer3$^{-/-}$ mice

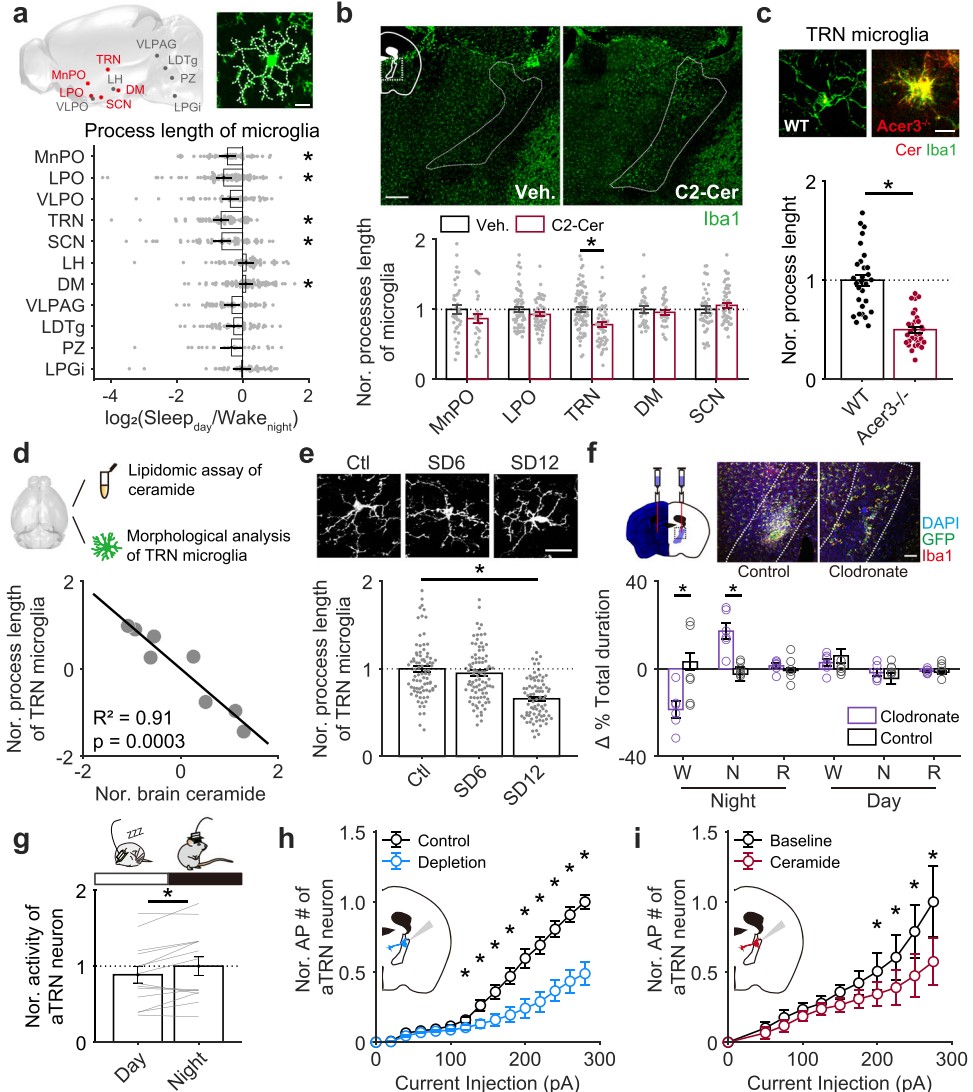

had higher Iba1 expression and shorter processes in the TRN region compared to those in wild-type mice (Fig. 4c, Supplementary Fig. 5a). To further validate the relationship between TRN microglia activity and brain ceramide level, we measure subcortical ceramide concentration and TRN microglial morphology from the same naïve mice and analyzed their correlation (Fig. 4d). We found that higher ceramide level was tightly correlated with shorter process length of TRN microglia (Fig. 4d). And TRN microglia dramatically retract their process in response to 12-hours of sleep deprivation (Fig. 4e), which is consistent with increased brain ceramide following the treatment (Fig. 2e). Taken together, these data suggest that TRN may be an essential region for microglia-mediated modulation of wakefulness stability.

To test this hypothesis, we locally depleted TRN microglia using focal clodronate liposome injections as shown in Fig. 4f and performed the same experiment and analyses as described in Fig. 1. Consistent with our findings for global microglia depletion, mice with microglia depleted locally in the anterior TRN (aTRN) displayed decreased wake durations at night and increased transitions between wakefulness and NREM sleep (Fig. 4f, Supplementary Fig. 5b).

We further hypothesized that microglia in the aTRN modulate wakefulness stability through influencing the neuronal activity via ceramide. To test this hypothesis, we first recorded aTRN

neuronal activity with tetrode recordings in mice during day and night times. We found that aTRN neurons displayed higher averaged firing rates at night than at day in normal mice (Fig. 4g). To test whether microglia depletion affects aTRN intrinsic excitability (such as firing capacity), we performed whole-cell patch recordings onto individual aTRN neurons in acute-prepared brain slices, by injecting various levels of currents and quantifying the elicited action potentials. We found that aTRN neuronal firing capacity was suppressed in microglia-depleted mice as compared to normal mice (Fig. 4h). In microglia-depleted mice, we have shown that night-time ceramide concentration was relatively elevated (Fig. 3b). Consistently, extracellular application of ceramide suppressed aTRN neuronal firing capacity from normal mice (Fig. 4i). Together these results indicated an inhibitory role of ceramide in modulating aTRN neuronal activity.

These data suggest that microglia may modulate stable wakefulness through ceramide signaling to aTRN neurons.

**aTRN neuronal activity regulated transitions between wake and NREM sleep.** To determine how aTRN neurons potentially modulate wakefulness stability, we first analyzed the night-time activity of aTRN neurons. We performed tetrode recordings in

**Fig. 4 High sensitivity of TRN microglia to brain ceramide. a** Diurnal differences in the total lengths of microglial processes in sleep/wakefulness-related brain regions (Sleep$_{day}$, 3 mice; Wake$_{night}$, 3 mice). A sagittal brain cartoon showing the regions examined, with red dots representing regions with significant Sleep$_{day}$/Wake$_{night}$ differences in microglial morphology. The white dashed line on the representative microglia indicates the measured processes. Scale bar, 10 μm. The $p$ value for each brain region was obtained by compare the difference of microglial process length between Sleep$_{day}$ and Wake$_{night}$, each dot represents log$_2$ of ratio (Sleep$_{day}$/mean value of Wake$_{night}$). MnPO, median preoptic area, $p = 0.015$ for Sleep$_{day}$ ($n = 65$) vs Wake$_{night}$ ($n = 49$); LPO lateral preoptic area, $p = 5.4 \times 10^{-4}$ for Sleep$_{day}$ ($n = 84$) vs Wake$_{night}$ ($n = 92$); VLPO ventrolateral preoptic area (Sleep$_{day}$, $n = 53$; Wake$_{night}$, $n = 55$); TRN thalamic reticular nucleus, $p = 1.5 \times 10^{-6}$ for Sleep$_{day}$ ($n = 64$) vs Wake$_{night}$ ($n = 70$); SCN suprachiasmatic nucleus, $p = 0.006$ for Sleep$_{day}$ ($n = 48$) vs Wake$_{night}$ ($n = 69$); LH lateral hypothalamus (Sleep$_{day}$, $n = 59$; Wake$_{night}$, $n = 70$); DM dorsomedial hypothalamus, $p = 0.039$ for Sleep$_{day}$ ($n = 83$) vs Wake$_{night}$ ($n = 88$); VLPAG ventrolateral periaqueductal grey matter (Sleep$_{day}$, $n = 64$; Wake$_{night}$, $n = 59$); LDTg laterodorsal tegmental nucleus (Sleep$_{day}$, $n = 89$; Wake$_{night}$, $n = 53$); PZ parafacial zone (Sleep$_{day}$, $n = 16$; Wake$_{night}$, $n = 25$); LPGi lateral paragigantocellular nucleus (Sleep$_{day}$, $n = 37$; Wake$_{night}$, $n = 48$). **b** Exogenous application of C2-ceramide specifically altered Iba1 fluorescence and microglial morphology by a retraction of processes in the TRN region (Veh. Vehicle, 6 mice; MnPO, $n = 38$ cells; LPO, $n = 25$ cells; TRN, $n = 85$ cells; DM, $n = 32$ cells; SCN, $n = 45$ cells. C2-Cer, C2-ceramide, 4 mice; MnPO, $n = 25$ cells; LPO, $n = 55$ cells; TRN, $n = 57$ cells; DM, $n = 31$ cells; SCN, $n = 51$ cells). $p = 2.5 \times 10^{-5}$ in TRN region for Veh. vs C2-Cer. Each dot represents one microglia. Brain tissue was collected 3 h after i.c.v. injection. Representative images with Iba1 staining were taken from TRN-containing brain sections, with the TRN outlined with white dashed lines. The dashed square within the cartoon inset indicates the brain regions from which images were taken. Scale bar, 200 μm. **c** Total process lengths of TRN microglia in wild-type (WT) and Acer3 knockout (Acer3$^{-/-}$) mice (WT, $n = 30$ cells from 2 mice; Acer3$^{-/-}$, $n = 31$ cells from 3 mice; $p = 8.1 \times 10^{-10}$ for WT vs Acer3$^{-/-}$). Each dot represents one microglia. Scale bar, 10 μm. **d** Correlation between brain ceramide level and microglial morphology in naïve mice. Subcortical ceramide level and total process length of TRN microglia were analyzed in the same mouse. Total process length of at least 20 TRN microglia was measured for each mouse, and the averaged value was used for further analysis. Pearson's correlation, $p$ value was calculated using two-sided hypothesis test. $n = 8$ mice. **e** TRN microglia retracted processes following sleep deprivation. Representative image with Iba1 staining shown TRN microglia in control and sleep-deprived mice. Scale bar, 20 μm. Ctl control mice, 86 cells from 4 mice; SD6, mice with 6 h of sleep deprivation, 91 cells from 4 mice; SD12, mice with 12 h of sleep deprivation, 81 cells from 4 mice. Each dot represents one microglia. **f** Local injection of clodronate liposomes into the anterior thalamic reticular nucleus (aTRN) effectively depleted microglia (top) and facilitated NREM sleep specifically at night (bottom). Scale bar, 100 μm. Clodronate vs Control at night: $p = 0.003$ for W; $p = 0.003$ for N. Clodronate liposomes, $n = 6$ mice; control liposomes, $n = 6$ mice. Each open circle represents the difference, before and after local administration, in one mouse, and then these differences were compared between clodronate-treated and control mice. **g** aTRN neuronal firing rate ($n = 14$ mice with 248 neurons been collected) during daytime and night-time. Each line represents change in mean firing rate of aTRN neurons between day and night in one mouse. $p = 0.012$ for day vs night. **h** Compared to control mice (11 neurons from 3 mice), action potentials induced by higher current injections were reduced in aTRN neurons from microglia-depleted mice (17 neurons from two mice; $p < 0.05$ for current injection high than 120 pA). **i** Application of C2-ceramide (10 μM) reduced aTRN neuronal excitability (9 neurons from 3 mice; $p < 0.05$ for current injection high than 200 pA). *$p < 0.05$; two-sided unpaired $t$-test for **a–c**, **e**, **f**, and **h**. Two-sided paired $t$-test for **g** and **i**. Data are reported as mean ± SEM. See also Supplementary Fig. 5.

the aTRN of normal mice together with EEG/EMG recordings, as shown in Fig. 5a and Supplementary Fig. 6a, and analyzed neuronal firing rates during different vigilance states. The tetrode data showed that 81% of aTRN neurons displayed higher firing rates during wakefulness than during NREM sleep (Fig. 5b). We then analyzed any changes in neuronal firing during the transition periods between these two states. We found that a large cohort of aTRN neurons dramatically decreased their firing rates from wake to NREM sleep, and increased their firing rates from NREM sleep to wake (Fig. 5c, d), indicating close synchronization to the onsets of these transitions. For 56–58% of these neurons, their firing rate changes within a 40-s window around transition onset were represented well by a logistic function (Fig. 5c, d; Supplementary Fig. 6b). We found that 75% of well-fit neurons exhibited firing rate decreases that preceded transitions from wake to NREM sleep (Fig. 5c, right panel), and that most neurons exhibited increased firing rates closely around the point of transition from NREM sleep to wake (Fig. 5d, right panel).

We then explored whether alterations of aTRN neuronal activity are sufficient to impact transitions from wake to NREM sleep. We employed a chemogenetic method to silence or activate aTRN neurons (Fig. 5e, f, Supplementary Fig. 8). To facilitate viral labeling specificity, we injected adeno-associated virus (AAV) carrying cre-dependent transgenes into GAD2$^{Cre}$ mice in which all GABAergic neurons expressed Cre recombinase[44]. Briefly, we bilaterally injected AAV-hsyn-DIO-KORD-mCitrine into the aTRN to express KORD (DIO-KORD), a kappa opioid receptor-based chemogenetic receptor (Fig. 5e, Supplementary Fig. 8b) that can be activated by salvinorin B (SalB)[45]. We verified that SalB suppressed the activity of KORD-expressing aTRN neurons (Supplementary Fig. 6c). To exclude the potential effect

of SalB injection per se, we prepared a group of GAD2$^{Cre}$ control mice with local expression of AAV-hSyn-DIO-mCherry in aTRN (DIO-mCherry). In mice expressing KORD receptors in aTRN neurons, SalB administration (compared to vehicle alone) dramatically increased the number of transitions between wake and NREM sleep, whereas administration of SalB in mice without KORD expression showed no change of transition (compared to vehicle injection) (Fig. 5e, Supplementary Fig. 7a). Consistently, suppression of aTRN neurons increased the bout numbers of wake and NREM sleep and decreased the bout durations of wake (Supplementary Fig. 7a,b). We should note that postmortem validations confirmed that viral expression was mainly in the aTRN (Supplementary Fig. 8b). We next chemogenetically activated aTRN neurons with AAV carrying hM3Dq (AAV-hSyn-DIO-hM3Dq-mCherry) (Fig. 5f, Supplementary Fig. 8c), which can be activated by clozapine-N-oxide (CNO)[46]. When we compared the sleep architecture of these mice between CNO-administration sessions and vehicle administration sessions, we found that the transitions between wake and NREM sleep decreased (Fig. 5f, Supplementary Fig. 7c), the bout numbers of wake and NREM sleep decreased and bout durations of wake increased (Supplementary Fig. 7c,d). There was no such change in control mice without hM3Dq expression that received the same CNO/vehicle injections (Fig. 5f, Supplementary Fig. 7c,d). Although there is a slight increase in bout durations of NREM sleep upon CNO injection in both mCherry-expressing and hM3Dq-expressing mice, no significant difference was observed between two mouse groups (Supplementary Fig. 7c,d). These results indicated that aTRN neuronal activity regulated the transitions between wake and NREM sleep, which in turn influence the stability of wakefulness. Combined with the finding

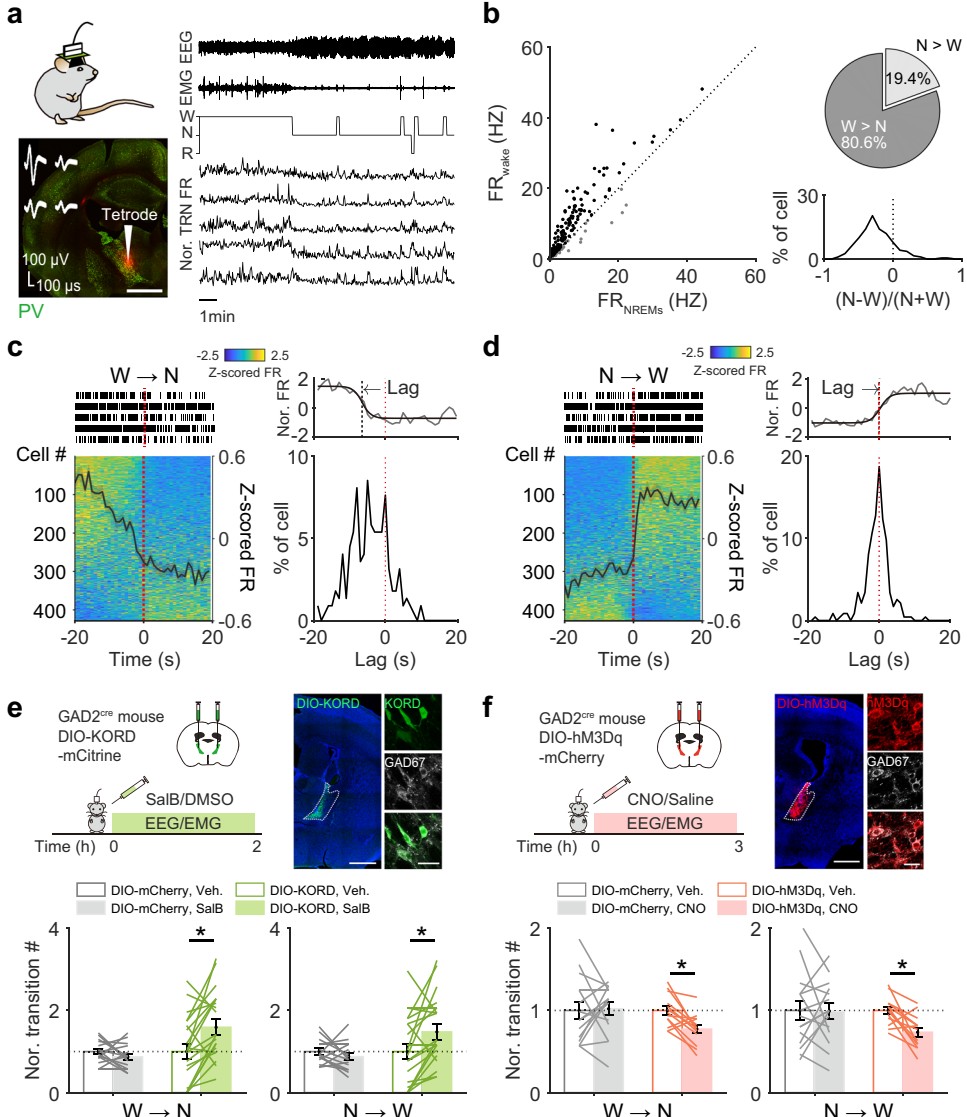

**Fig. 5 aTRN neuronal activity controls transitions between wakefulness and NREM sleep. a** Simultaneous recordings of aTRN neuronal activity and sleep architecture via tetrode/EEG/EMG implants. The TRN is represented with parvalbumin (PV) staining (red signal, tetrode tip location). **b** Left, scatter plot of aTRN neuron firing rates (FR) during wakefulness (y-axis) against during NREM sleep (x-axis). Right, distribution of firing preferences between wakefulness and NREM sleep (191 neurons from 14 mice). **c, d** Left side, activity of aTRN neurons 20 s before and after transition onsets from wakefulness to NREM sleep (**c**, W → N), or **d** from NREM sleep to wakefulness (N → W, 428 cells from 23 mice). The black trace indicates average normalized neuronal activity. The raster plots represent spike events from five representative aTRN neurons. Right side, lag time distributions relative to transition onset (the red-dashed line), and an example of logistic fitting is shown at the top. **e, f** Chemogenetic inhibition (**e** via DIO-KORD), or activation (**f** via DIO-hM3Dq), of aTRN neuron controls state transitions between wakefulness and NREM sleep. Top, experimental procedure (left) and validation (right) of virus expression in the aTRN of GAD2$^{cre}$ mice (scale bars: left = 1 mm; right = 50 μm). Bottom, differences in state transition numbers between wakefulness and NREM sleep in mice with chemogenetic inhibition (**e**, DIO-KORD, $p = 0.002$ for W → N and $p = 0.010$ for N → W, $n = 21$ pairs of recordings from 7 mice; DIO-mCherry, $n = 15$ pairs of recordings from 5 mice) or with chemogenetic activation of aTRN neurons (**f**, DIO-hM3Dq, $p = 0.004$ for W → N and $p = 0.002$ for N → W, $n = 13$ pairs of recordings from 4 mice; DIO-mCherry, $n = 15$ pairs of recordings from 5 mice). Each line represents the change in transition number between recording sessions with vehicle or CNO/SalB injection collected from one mouse. *$p < 0.05$; two-sided paired Wilcoxon signed rank test for **e** and **f**. Data are reported as mean ± SEM. See also Supplementary Figs. 6–8.

in microglia-depleted mice, our data suggest that microglia may modulate wakefulness stability via regulation of aTRN neuronal activity.

## Microglial depletion decreased aTRN neuronal activity during wakefulness.

Microglia have been shown to regulate neuronal activity under both physiological and pathological conditions[47–50]. To examine whether impaired wakefulness after microglial

depletion resulted from changes in aTRN neuronal activity, we used simultaneous tetrode and EEG/EMG recordings to monitor aTRN neuronal activity under different vigilance states before and after microglial depletion (Fig. 6a). In microglia-depleted mice, aTRN neuronal firing rates during wake were significantly lower on day 4 compared to day 0 (from $10.2 ± 1.71$ spikes/s to $6.9 ± 1.7$ spikes/s, $p = 0.036$) (Fig. 6b), whereas during NREM sleep, firing rates exhibited little change from day 0 to day 4 (from $6.3 ± 1.3$ spikes/s to $4.8 ± 1.5$ spikes/s, $p = 0.093$). As a control, we performed the

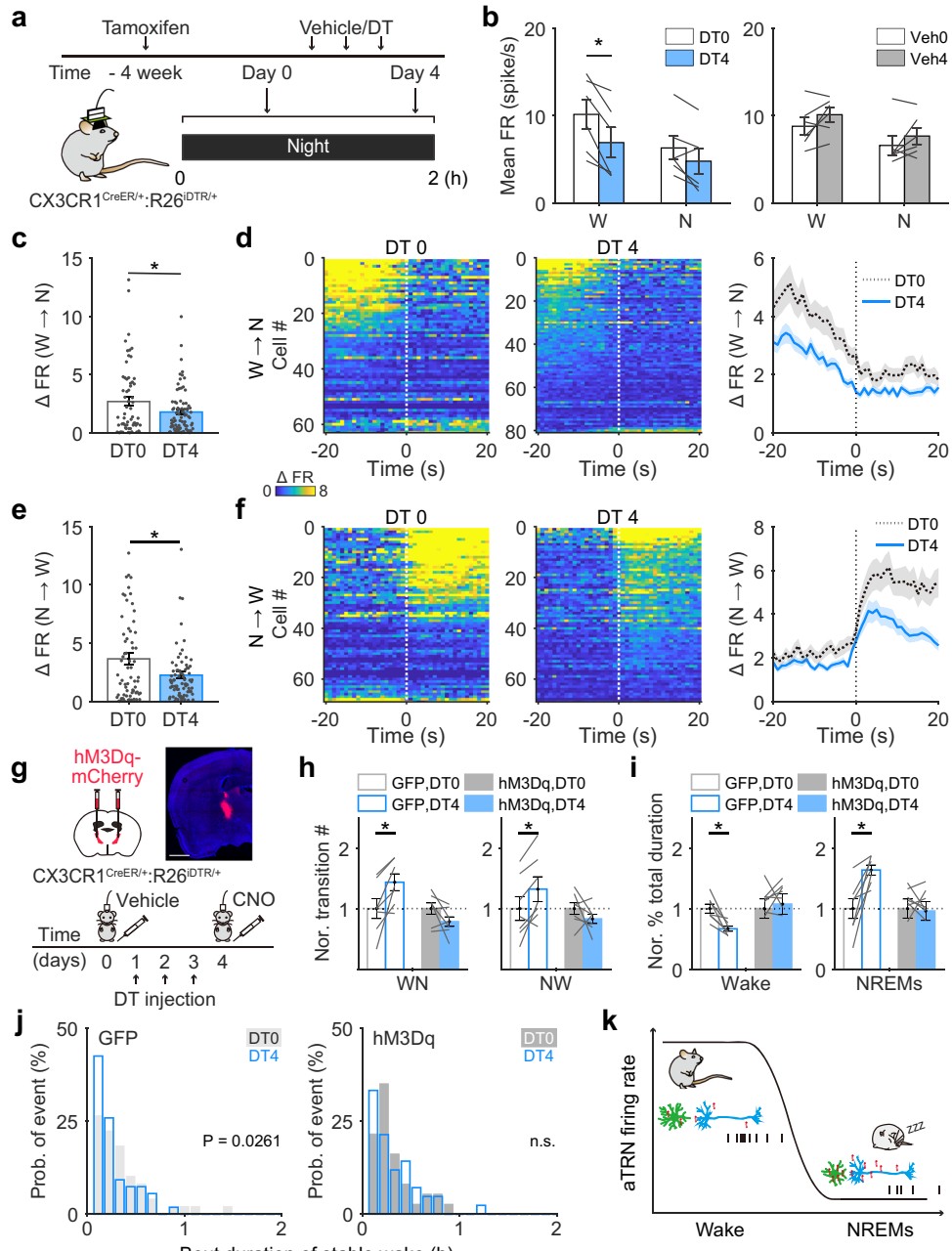

same set of experiments in mice with the same genetic background and saline administration instead of DT, and found no changes between day 0 and day 4 during either wake or NREM sleep (Fig. 6b). Importantly, firing rate differences between wake and NREM sleep decreased after microglia depletion (Fig. 6c–f), suggesting that the closer activity levels of aTRN neurons during these two states may have accounted for the easier transitions between them. We should note that there were no detectable changes in the proportion of neurons that exhibited logistic firing rate or lag time changes during these state transitions in microglia-depleted mice (Supplementary Fig. 9a,b). The same set of analyses were performed in control mice without microglia depletion, and no changes were observed (Supplementary Fig. 9c–d).

**Chemogenetic activation of aTRN neurons restored the microglial depletion-induced decrease in wakefulness stability.** Our findings that aTRN neuronal activity regulated transitions between wake and NREM sleep, and that microglial depletion

increased transitions between wake and NREM sleep and decreased aTRN neuronal activity, support the hypothesis that microglia modulate the the transitions through influencing aTRN neuronal activity. Thus, we may be able to restore normalcy of state transitions in microglia-depleted mice by increasing aTRN neuronal activity. We used chemogenetic activation of aTRN neurons in microglia-depleted mice (Fig. 6g). Briefly, 1 week after tamoxifen administration in CX3CR1^Cre-ERT2/+:R26^iDTR/+ mice (this induced DT receptor expression in microglia), we injected AAV-hSyn-hM3Dq-mCherry into the aTRN. Another group of CX3CR1^Cre-ERT2/+:R26^iDTR/+ mice with tamoxifen induction and AAV-CAG-GFP injections served as controls. Three weeks after viral injection, we used CNO (i.p.) to activate hM3Dq-expressing aTRN neurons[46] following microglial depletion. We performed EEG/EMG recordings to monitor the sleep architecture in these mice within a time window of 3 h after vehicle/CNO injection (Fig. 6g). We found that chemogenetic activation of aTRN neurons in microglia-depleted mice indeed restored the

**Fig. 6 Attenuation of aTRN neuronal excitability is responsible for changes in wakefulness stability following microglial depletion. a** aTRN neuronal activity and sleep architecture were recorded at night, before and after microglial depletion. **b** Averaged firing rates (FR) of aTRN neurons before and after DT or vehicle administration ($n = 6$ DT-injected mice, $n = 6$ vehicle-injected mice). Each line represents change in mean firing rate of aTRN neuron from one mouse. W wakeful, N NREM sleep. DT 0 vs DT 4, $p = 0.007$ for wake state. **c, e** Differences in neuronal firing rates between wakefulness and NREM sleep during transitions from wakefulness to NREM sleep (**c** DT 0, $n = 63$ neurons from 6 mice; DT 4, $n = 81$ neurons from 6 mice; $p = 0.028$ for DT 0 vs DT 4) and from NREM sleep to wakefulness (**e** DT 0, $n = 69$ neurons from 6 mice; DT 4, $n = 81$ neuron from 6 mice s; $p = 0.017$ for DT 0 vs DT 4) before/ after microglial depletion. Spike events occurring 5–20 s before/after transition onsets were included. **d, f** aTRN activity during transitions from wakefulness to NREM sleep (**d**) and from NREM sleep to wakefulness (**f**) before/after microglial depletion. Neuronal activities within ±20 s from transition onsets were normalized to their minimum firing rates. Right panels, averaged aTRN neuronal activities during transitions. **g** Experimental procedure for chemogenetic activation of aTRN in microglia-depleted mice. Top, validation of hM3Dq expression. Scale bar = 1 mm. **h, i** The number of state transition (**h** left: WN wakefulness to NREM sleep; right: NW NREM sleep to wakefulness; GFP, DT 0 vs DT 4: $p = 0.015$ for WN, $p = 0.020$ for NW) and % total duration of wake and NREM sleep (**i** left, total duration of wake; right, total duration of NREM sleep; GFP, DT 0 vs DT 4: $p = 0.005$ for wake, $p = 0.007$ for NREM sleep) were compared before/after microglial depletion in GFP-expressing ($n = 7$) or hM3Dq-expressing ($n = 7$) mice. Each line represents change in transition number from one mouse. **j** Left panel: in control mice expressing GFP, microglial depletion significantly shortened the bout durations of stable wakefulness ($n = 7$ mice). Right panel: chemogenetic activation of the aTRN abolished the effect of microglial depletion on wakefulness stability ($n = 7$ mice). **k** A proposed model for ceramide-mediated microglia-neuron interactions in wakefulness/NREM sleep regulation. Microglia regulate the diurnal variation in ceramide levels, which determines the neuronal activity of the aTRN and, in turn, controls transition states between wakefulness and NREM sleep. Green cells: aTRN microglia; blue cells: aTRN neurons; red molecules: ceramide; black raster plot, action potentials produced by aTRN neurons. *$p < 0.05$; two-sided unpaired t-tests for **c** and **e**; two-sided paired t-tests for **b, h**, and **i**. two-sided Kolmogorov–Smirnov test for **j**. Data are reported as mean ± SEM. See also Supplementary Fig. 9.

transition numbers between wake and NREM sleep, percentage of total time for wake, and wakefulness stability (Fig. 6h–j, Supplementary Fig. 9e).

In summary, our findings indicated that microglia could modulate wakefulness stability through ceramide-mediated interactions between microglia and neurons in the aTRN (Fig. 6k).

## Discussion

The mechanisms involved in regulating vigilance state of brain expand in multiple systematic layers, from fine-turned neural circuits controlling to circadian rhythm regulation, from internal metabolic dynamics to external environment stimuli[16,51–53]. Those mechanisms function in a wide-range timescale, with neurons synchronize their activity with brain state transitions in milliseconds[54–56] whereas microglia survey the brain and sense the behavioral alterations in minutes[6,38].

In this study, we depleted microglia using the CX3CR1[CreERT2/+]: R26[iDTR/+] mice. The Cre expression in CX3CR1-expressing cells is induced by tamoxifen treatment, which in turn turns on DT receptor expression in CX3CR1-expressing cells[27,28]. In addition to microglia, some other peripheral and circulating immune cells also express CX3CR1[29,30,57]. Thus, the wakefulness change observed in our study may include contribution of cells other than microglia. However, we believe it is less likely for the following evidences. First, we confirmed that in the brain parenchyma, the CX3CR1-expressing cells are microglia (Fig. 1c and Supplementary Fig. 1c), consistent with previous report[27,28,31,38,50,58]. Second, taking the advantage that peripheral CR3CR1-expressing cells have much higher turnover rates than microglia (<2 weeks versus >8 weeks), we perform the DT injection 4 weeks after the tamoxifen treatment to specifically deplete microglia, with no detectable effect on peripheral CX3CR1-expressing cells (Fig. 1b). Third, DT receptor mediated microglia depletion do not compromise the blood-brain barrier[28], exclude the possibility of circulating immune cells infiltrating the brain. We also confirm that within the time window of our analyses, there is no change in densities of neuron and astrocyte (Supplementary Fig. 1 d–f).

It would be ideal to quantify the correlation of metabolites and microglia morphology directly with wake/sleep instead of day/ night in mice since mouse sleep/wake cycle is less homogenous than the human[13]. However, it is technically challenging to

monitor metabolite concentrations and/or microglia processes in free-moving mice with EEG recordings to identify wake/sleep epochs. Currently ceramide sensor has poor specificity and limited availability[59]. Two-photon imaging that can monitor the microglia processes required head-fixed configuration of animals[6,7], whereas single-photon imaging that can be coupled on freely moving animals is not sensitive enough to identify the thin processes of microglia. Therefore, in this study we addressed this question in an alternative strategy. First, the brain samples we collected for metabolite and microglia analyses (Figs. 2–4) were (1) all from sleeping mice at day or awake mice at night, and (2) at time points mice having 65% sleeping time or 25% sleeping time, respectively (Supplementary Fig. 4d). Thus, although mouse sleep/wake cycle is not exactly synchronized with day/night, the samples were collected from mice with big differences in sleep and wake periods. Second, we monitored the mouse behaviors through infrared video recording for 1 h before sampling. We found the percentage of wake periods in the last 1 h before sample collection is highly correlated with the brain ceramide concentration (Fig. 2f), and the brain ceramide concentration is highly correlated with the total length of TRN microglia processes (Fig. 4d). Together, we believe our analyses reveal the sleep/wake related correlations in metabolite concentration and microglia processes.

Previous studies associated microglia with sleep in two lines of evidence. First, sleep deprivation activates microglia[10,11,60,61] and microglia morphology and activity change in different brain vigilance states, such as being awake or under anesthesia[6,7]. Second, sleep is impaired in diseases accompanied with microglial activation, such as Alzheimer's disease and parasitic infections[8,9]. Our current study provides the evidence for microglial function in modulating stable wakefulness. The finding that microglial depletion decreases the duration of night-time wakefulness due to an increase in transitions from wake to NREM sleep (Fig. 1) indicates that microglia are important for maintaining the stability of wakefulness. It should be noted that we observed these changes only during night-time (Fig. 1d, Supplementary Fig. 3a–c). One reason for this difference between day and night observations is that mice sleep mostly during the day, and the few stable bouts of wake during the day may limit our ability to detect changes. However, it is also possible that sleep (and/or transitions between wakefulness and NREM sleep) is regulated differently

between day and night. Further studies, especially those using tools and paradigms for in vivo monitoring and manipulation of microglial activity, are needed to examine these possibilities. Microglial quickly repopulated after depletion, during which astrogliosis may observed[28,62–64]. However these alterations do not occur right after microglia depletion[27,28], which is the time frame of our study. As repopulated microglia can promote brain recovery after injury[65], benefit neurological disorder treatments[66], and improve cognitive function in aging mice[67,68], further studies are also needed to understand whether microglial repopulation affects stable wakefulness.

In the current study we showed that aTRN regulates the transitions between wake and NREM sleep. The sharp changes in firing rates of aTRN neurons precede transition onsets from wake to NREM sleep, but not from NREM sleep to wake, suggesting that aTRN neuronal activity may only drive transitions from wake to NREM sleep. Previous studies showed that optogenetic silencing of TRN led to a rapid arousal from NREM sleep[69,70], which is consistent with our findings that chemogenetic silencing of aTRN increases the number of transitions from NREM sleep to wake (Fig. 5e, Supplementary Fig. 7a–b). However, it is unclear whether previous findings match our findings of increased transitions from wake to NREM sleep. It is probable that our study focuses on different portions of the TRN compared to previous studies: limbic TRN vs sensory TRN. The TRN is a heterogeneous brain region with high molecular, cellular, anatomical, and functional diversity[19,24]. From anterior to posterior, the TRN can be divided into different regions based on target projections, such as the limbic TRN and sensory TRN[19]. TRN neurons also exhibit distinct firing patterns from anterior to posterior; the limbic TRN (aTRN) is composed mainly of wakeful/REM sleep-active neurons, whereas activity in the sensory TRN (posterior TRN) is either state-independent or sleep-active[17,20,25,26]. Most of the neurons that we recorded from in this study exhibit high firing rates when mice were awake and low firing rates when mice were in NREM sleep (Fig. 5a–d), and postmortem analyses confirmed recording locations in the aTRN (Supplementary Fig. 6a). There is a growing body of evidence suggesting that anterior part of the TRN is implicated in selective attention, fear conditioning, and flight behavior[20,22,23,71], for which stable wakefulness is a prerequisite. Sensory TRN exhibits strong repetitive burst discharges[24,72], which is essential for generating spindle oscillations and in turn promotes sleep onset[41,73–75]. A recent study further dissected the contributions of subpopulations of sensory TRN neurons to spindle activity and NREM sleep-bout duration[24]. Combining previous reports and our current findings, it is possibly that limbic TRN activity is essential for wakefulness stability and sensory TRN activity is crucial for NREM sleep stability. Whether and how these two regions of the TRN interact in wakefulness and sleep regulation requires further study.

The aTRN neurons send projections to intermediodorsal, anterodorsal, and dorsal midline thalamus[71,76]. Those connections provide the structure basis for function of aTRN in in selective attention, fear conditioning, and flight behavior[20,22,23,71], for which stable wakefulness is a prerequisite. Based on the coordinates in this study, the recording and manipulations covered most of the aTRN (Supplementary Fig. 6a, 8). It is not clear whether all aTRN neurons or a subpopulation of aTRN neurons are essential for the phenotypes we observed. Furthermore, the aTRN mainly contains limbic and motor sector of TRN area[19,24], receiving projections from relevant thalamic nuclei (e.g., anterdorsal and laterodorsal thalamus) and cortical areas (e.g., prelimbic cortex and cingulate cortex)[22,76]. The specific inputs and outputs of the aTRN that controlling wakefulness stability and the transition between wakefulness and NREM sleep, are of great interest for future studies.

As the two main cell populations in the brain, there is a reciprocal interaction between microglia and neuron[49,50,77]. In this study, we show that elevation of ceramide following microglial depletion decreased intrinsic excitabilities of aTRN neurons (Figs. 4h, i, 6a–f) and manipulations of ceramide productions either mimicked or blocked these microglia depletion effects (Figs. 3c, d, 6g–j), suggesting that microglia can modulate neuronal activity through ceramide. Ceramide locates in the branching point of sphingolipids metabolism[78], and most of brain cells may be able to produce ceramides. Ceramide was previously found to affect microglia activity levels and neuronal activity[79–81]. It remains unclear whether the ceramide critical for aTRN neuronal activity is produced from microglial cells or other cell types (e.g., astrocyte). Furthermore, whether and how ceramide directly interacts with neurons are largely unknown. Sphingosine-1-phosphate (S1P) is a main catabolite of ceramide[82]. A recent work from our group revealed that sphingosine-1-phosphate receptor 2 (S1PR2) is specifically located in interneuron, with remarkable expression in interneuron enriched TRN region[83]. Manipulations of S1P signaling via S1PR2 tune inhibition level in the neural network[83]. Therefore, it is possible that microglia modulate aTRN neurons via ceramide-s1p-S1PR2 signaling pathway. Although we have shown that within the study time window the cell density of astrocytes was not changed (Supplementary Fig. 1d & e), it did not rule out the possibility that astrocytes and other brain cells are also involved in the modulatory process. The molecular and cellular mechanisms underlying how microglia regulate subcortical ceramide concentrations, and how ceramide modulates neuronal excitability require further studies.

Our study does not rule out the potential involvement of other signaling molecules, such as adenosine. Adenosine-related metabolites are essential modulators of microglial morphology and activity[84–86]. A recent study reported that adenosine produced by microglia is crucial for the suppressive role of microglia on neuronal activity[49]. Adenosine maintains a low level during NREM sleep, and accumulates during wake and within REMs[52,87]. Manipulation of either brain adenosine level or its receptors, especially in the basal forebrain, produced a prominent effect on sleep/wake behavior[52,87,88]. Thus, it is highly possible that adenosine may also involve the microglia modulation of stable wakefulness, by functioning in either parallel or joint pathways with ceramide signaling. Previous studies provided limited evidence about the relationship between ceramide and adenosine. Ceramide acts as a pro-inflammation factor[89], whereas adenosine is an anti-inflammation factor[90]. Ceramide may affect adenosine production by blocking mitochondrial ATP release[91]. Adenosine suppresses tumor necrosis factor-induced nuclear factor- κB (NF-κB) activation but has less effect on ceramide-induced NF-κB activation[92]. Adenosine executes a strong suppression on excitatory synaptic transmission, but not on inhibitory synaptic transmission[93]. Future studies may reveal whether and how adenosine (or other signaling molecules) participates in this microglia-mediated wakefulness modulation.

Brain ceramide levels vary across different physiological and pathological conditions. Long chain ceramides are dramatically increased in aged brains[43,94]. Similarly, ceramide levels have been associated with neurodegenerative disease severity, such as in Alzheimer's disease[94–96]. With aging (in both rodents and humans), and in patients with neurodegenerative diseases, there are deficits in wakefulness stability[97–100]. The present findings that microglia regulate wakefulness stability through ceramide may shed light on potential mechanisms underlying the impaired wakefulness with aging and in other pathological conditions.

## Methods

**Animals**. Male and female adult (2–5 months old) wild-type C57BL/6 J, CX3CR1[CreERT2/+], CX3CR1[CreERT2/+]:R26[iDTR/+], CX3CR1[CreERT2/+]:R26[iDTR/−], GAD2[Cre] mice, and Acer3[−/−] mice were used for the experiments. Mice were entrained to a 12 h/12 h light/dark cycle with 64–79 F and 30–70% humidity, food and water available ad libitum. All procedures were approved by the Stony Brook Animal Care and Use Committee and carried out in accordance with the National Institutes of Health standards.

**Surgery**. Adult mice were anesthetized with isoflurane (3% for induction followed by 1% for maintenance) and placed in a stereotaxic apparatus. Both eyes were covered with a layer of ointment during surgery. After shaving and sterilizing the skin, the scalp was incised to expose the skull. Connective tissue was gently removed with a scalpel, and a thin layer of iBond Self Etch was applied.

For EEG/EMG electrode implantation, four holes were drilled into the skull at the following coordinates: 1st EEG channel: AP:+2 mm, ML:+1.5 mm; 2nd EEG channel: AP: −4 mm, ML:+1.5 mm, reference channel (AP:+2 mm, ML: −1.5 mm), and ground channel site (AP: −4 mm, ML: −1.5 mm). A 2-EEG/1-EMG headmount (8201, Pinnacle Technology Inc.) was aligned to the craniotomy sites and prefixed to the skull with a small drop of dental cement. Four EEG screws (0.10 in EEG screws for the anterior craniotomy sites and 0.12 in EEG screws for the posterior craniotomy sites) were inserted into the corresponding channels of the headmount and screwed to the skull. Two EMG leads were inserted on each side of the neck muscles to record postural tone. All implants were secured to the skull with dental cement.

For tetrode/EEG/EMG implantations, three holes were drilled into the skull at the following coordinates: EEG channel: AP: −4 mm, ML: −1.5 mm; reference channel, AP:+2 mm, ML:+1.5 mm; ground channel, AP: −6 mm, ML: 0 mm. Screw electrodes were soldered with PTFE-coated silver wire (0.005 in, WPI) and screwed into each craniotomy site, and two EMG wires were inserted on each side of the neck muscles. To record aTRN neuronal activity simultaneously, one additional craniotomy was created over the aTRN (AP: −0.58 mm, ML:+1.25 mm). The tetrodes were assembled as described previously[101]. Briefly, each tetrode consisted of four polyimide-coated nichrome wires (12.7 µm wire diameter; Sandvik, Palm Coast, FL, USA) twisted together, placed in a polyimide tube (ID, 0.0049 in; OD, 0.0064 in, Cole-Parmer) and gold-plated to an impedance of 0.3–0.5 MΩ at 1 kHz. One end of the tetrode assembly was glued to a microdrive to ensure vertical movement of the entire apparatus. Seven customized tetrodes along with two EEG wires, two EMG wires, one reference wire, and one ground wire were anchored to an electrode interface board (EIB-32 narrow, Neuralynx) with gold EIB pins (Neuralynx). The tips of the tetrodes were brushed with DiI (V22885, Vybrant) for postmortem identification of the recording site (Supplementary Fig. 6a). The dura above the aTRN was removed, the tip of the tetrode was placed over the target area, and then it was slowly lowered to 2.8 mm below the cortical surface. Three percent agarose was applied to the exposed tetrode wire and brain tissue to protect the area from the dental cement. The implants were secured to the skull with dental cement.

For cannula/EEG/EMG implantations, two holes were drilled over the bilateral aTRN (AP: −0.58 mm, ML:±1.25 mm). Custom-designed guide cannulae (3.5 mm, C315GMN, Plastic One) were slowly lowered to 2.5 mm below the cortical surface and secured with a small drop of dental cement. Four additional holes were drilled into the skull at the following coordinates: 1st EEG channel: AP:+2 mm, ML:+1.5 mm; 2nd EEG channel: AP: −4 mm, ML:+1.5 mm, reference channel (AP:+2 mm, ML: −1.5 mm), and ground channel site (AP: −4 mm, ML: −1.5 mm). Screw electrodes were soldered with PTFE-coated silver wire (0.005 in, WPI) and screwed into each craniotomy site. Silver wires were soldered to the corresponding channels of the 2-EEG/1-EMG headmount. All implants were re-secured with dental cement. Dummy cannulae (3.5 mm, C315DCMN, Plastic One) were mounted to the implanted guide cannulae after the dental cement had dried.

For virus injections, two holes were drilled into the skull bilaterally over the aTRN (AP: −0.58 mm, ML:±1.25 mm). A homemade glass micropipette (10–15 µm tip diameter) was inserted from the surface of the brain to 3 mm below the cortical surface. The virus (300 nl) was injected via glass pipettes connected to a Picospritzer II microinjection system (Parker Hannifin Corporation). The glass pipette was left in position for at least 5 min, and then slowly withdrawn. The virus used in the current experiments was obtained from AddGene. The virus and mouse line assignments are listed in Supplementary Table 2.

For i.c.v. injections of ceramide, a craniotomy was made over the lateral ventricle (AP: −0.4 mm; ML: 1.1 mm), and 5 µl of C2-ceramide (293 µM) or vehicle (1% ethyl alcohol in saline) was delivered via glass pipette 2 mm below the cortical surface as described previously.

Following surgery, mice were kept on a heating pad and closely monitored until they were fully awake, and then housed individually post-surgery. Mice were allowed to recover for at least 2 weeks.

**In vivo electrophysiological recordings and data analysis**. For all in vivo recordings, mice were habituated to the tethered recording system in their open-top home cage for 2 h per day for at least 3 days. Food and water were available ad libitum. Recordings were performed in the animals' home cages to reduce their

stress levels and minimize their exploration in a novel environment. All cage maintenance work was performed after each daily recording session was completed. Mice were able to move freely around their cages during the in vivo electrophysiology experiments. For recordings that included repeated i.p. injections (SalB, CNO, and carmofur), the interval between injections was strictly controlled to ≥3 days to reduce the potential effect of injection-induced stress on sleep architecture.

Mice with the 2-EEG/1-EMG headmount implants were tethered to a preamplifier (8202-SL, Pinnacle Technology, Inc.) and a commutator (8204, Pinnacle Technology, Inc.) for free-moving recordings. Signals were collected with a 100× gain, 0.5 Hz high-pass filter and 400 Hz sampling rate using Sirenia Acquisition software (Pinnacle Technology, Inc.) and analyzed with Sirena sleep software (Pinnacle Technology, Inc.). Simultaneous video recording was performed to monitor animal behavior. Brain state scoring was first processed using the cluster scoring feature of Sirenia Sleep, and then manually reinspected epoch-by-epoch based on the frequency and amplitude signatures of the signals, and corresponding behaviors on video. States were assigned based on consecutive, nonoverlapping, 4-s windows as either wakeful, NREM sleep or REM sleep using scoring criteria described previously[13,15]. Briefly, low-amplitude desynchronized EEG and high EMG activity were defined as wakeful; synchronized EEG with high-amplitude low-frequency (0.5–4 Hz) EEG activity and low EMG activity were defined as NREM sleep; and prominent theta frequency (6–9 Hz) EEG and low EMG activity were defined as REM sleep. Proportions (%) for total duration, bout duration, and bout number/h for each brain state were calculated. A state transition was defined as a change in brain state lasting for more than 12 s. Due to innate properties of mouse sleep[13], only state transitions from wakefulness to NREM sleep, from NREM sleep to REM sleep, from NREM sleep to wakefulness, and from REM sleep to wakefulness were calculated. As wakeful-state bout durations varies within a wide range in mice[102], we defined wakeful states as either quick arousals (short wakeful bouts lasting more than 12 s, followed by a transition back to sleep) or stable wakefulness (long wakeful bouts for essential behavior, such as food intake and exploration) based on the duration of each wakeful episode; wakeful bouts less than 1 min were defined as quick arousals, and wakeful bouts longer than 2 min were defined as stable wakefulness.

For animals assigned to the microglial depletion experiments (both global and local depletions), recordings were performed during the middle of the day (Zeitgeber time [ZT] 3–9) and night (ZT 15–21). For global depletions, mice were recorded before and after three days of DT/vehicle injections. For local depletions, mice were recorded 6 h after clodronate/control liposome injections. Differences before and after injections were calculated for each animal and compared between groups. For GAD2[Cre] mice with DIO-hM3Dq/control virus expression and CX3CR1[CreER/+]:R26[iDTR/+] mice with hM3Dq/control virus expression, 3 h recordings were performed at night following CNO or vehicle (saline) i.p. injections. For mice with DIO-KORD/control virus expression, 2 h recordings were performed at night following SalB or vehicle (DMSO) s.c. injections. For mice with carmofur injections, signals were collected at night for 4 h following injections. For the GW-4869-mediated rescue experiments in microglia-depleted mice, signals were recorded at night for 6 h following GW-4869/vehicle injections. The start time of the recording was fixed for each animal. Differences between vehicle and drug injections were compared for each animal.

For mice with EEG/EMG/tetrode implantations, we simultaneously recorded EEG, EMG, and aTRN neuronal activity across spontaneous sleep-wakeful states in freely moving mice for 2 h at night. Implanted mice were tethered to a preamplifier (HS-36-LED, Neuralynx), which was connected with a long cable hanging from the ceiling of the recording chamber to allow for free-moving recordings. To balance the weight of the implants and preamplifier on the heads of tethered mice, a lever was used with one end tethered to the preamplifier using a cotton thread, and the other end secured with a similarly weighted metal screw. Signals were filtered (0.5–100 Hz for EEG/EMG signal, 600–6000 Hz for spike activity) and sampled at 40,000 Hz using the Neuralynx 32-channel hybrid system and Cheetah data acquisition software (Neuralynx). Single units were isolated offline using the MATLAB-based function MClust (The Mathworks, Natick, MA, USA). All isolated units with stable firing during recordings were retained. A peak-to-trough signal less than 200 µs was identified as a putative TRN interneuron[20,103], and further confirmed during postmortem analyses (Supplementary Fig. 6a). EEG/EMG signals were downsampled to 400 Hz and scored in 4-s epochs using customized MATLAB code based on the criteria described above. To compare aTRN neural activity between day and night, global activity (total spike events during the recording period divided by the total time of the recording) was calculated. The neuronal activity of identified units was aligned with the sleep architecture for each recording. The averaged neural activity for different brain states was calculated separately as the total number of spikes during a certain state divided by the total time (in seconds) spent in that state. Neuronal activity during the transition period was defined as spikes that occurred within 20 s before and after the onset of state transitions. State transitions that satisfied the following criteria were included in further analyses of neuronal activity during the transition period: (1) the prior and subsequent state lasted >20 s; and (2) the analyzed transition behavior occurred three times or more during the recording period. Neuron transitional activity was defined as the averaged firing rate of a neuron during periods of the same transition behavior across a single recording. Neuron transitional activity was fitted by the

following logistic function:

$$y = \min + \frac{\max - \min}{1 + 10^{-a \times (x - LT)}} \qquad (1)$$

where y is the neuronal activity, $x$ is the time relative to the onset of the state transition, max and min represent maximal and minimal values of the fitted curve, a is slope, and $LT$ is the lag time.

For each neuron, the quality of the fit was assessed using the following equation

$$R^2 = \frac{\sum_i (f_i - \bar{y})^2}{\sum_i (y_i - \bar{y})^2} \qquad (2)$$

where $f_i$ represents the fitted neuronal activity, $y_i$ represents the raw neuronal activity, and $\bar{y}$ is the averaged activity of this unit. Units with $R^2 > 0.3$ were retained for the calculation of the lag time.

Tetrodes were lowered by 50 μm per day to find the optimal position from which to maximally identify cells within the aTRN. Once the optimal position was found, the tetrodes were kept in this position until all the recordings for that animal were completed. The start time of the recording was fixed for each animal.

**Drug applications**. We used a CX3CR1[CreERT2/+]:R26[iDTR/+] transgenic mouse line for global microglial depletions[27,28]. Mice were administered tamoxifen (T5648, Sigma) to initiate DT receptor expression (500 mg/kg, gavage twice, with 2-day intervals) at 4 weeks before depletion onset. Microglia labeled with DT receptor were depleted following i.p. injections of DT (D0564, Sigma) for 3 consecutive days (25 μg/kg). For local microglia depletions, 1 μl of clodronate liposomes or control liposomes (SKU# CLD-8901, Encapsula Nano Sciences) was applied bilaterally to the aTRN using an injection cannula (4 mm, C315IMN, Plastic One) fixed to cannula-implanted mice.

For the chemogenetic activation experiments, CNO (BML-NS105, Enzo) was dissolved in saline and i.p. injected (0.3 mg/kg) into GAD2[Cre] mice expressing DIO-KORD/DIO-mCherry, or CX3CR1[CreER/+]:R26[iDTR/+] mice expressing hM3Dq/GFP; saline served as the control. For the chemogenetic inhibition experiments, SalB (HB4887, HelloBio) was dissolved in DMSO and injected s.c. (10 mg/kg); DMSO served as the vehicle.

The acid ceramidase inhibitor carmofur (Item No. 14243, Cayman) was dissolved in 15% Tween-80, 15% polyethylene glycol, and 70% saline, and then i.p. injected into wild-type mice (30 mg/kg); 15% Tween-80, 15% polyethylene glycol, and 70% saline without carmofur served as the vehicle. The SMase2 inhibitor GW-4869 (Item No. 13127, Cayman) was dissolved in DMSO (5 mg/ml) and then diluted with saline for i.p. injection (0.625 mg/kg) into CX3CR1[CreER/+]:R26[iDTR/+] mice before (three times, with 3-day intervals) and after microglial depletion; 2.5% DMSO served as the vehicle.

**In vitro electrophysiological recordings**. Mice were deeply anesthetized with urethane (2.5 g/kg, i.p.) and transcardially perfused with precooled NMDG artificial cerebrospinal fluid (ACSF). Brains were quickly dissected on ice and sliced (280 μm thickness) with a vibratome (HM650V, Thermo Scientific). Coronal sections of the aTRN were kept in HEPES ACSF. Whole-cell recordings were performed in 32 °C, oxygenated, recording ACSF under current-clamp conditions. Signals were filtered at 2.9 kHz and digitized at 10 kHz with a HEKA EPC-10 amplifier and PatchMaster software (HEKA Electronics, Lambrecht (Pfalz), Germany). Data analysis was conducted using Clampfit 10.0 (Axon, Sunnyvale, CA, USA). To test the excitability of aTRN neurons in microglia-depleted mice and aTRN neurons exposed to 10 μM C2-ceramide (62510, Cayman), stepped currents were injected into the patched neurons, and the number of induced action potentials was calculated.

The NMDG ACSF contained (in mM): 93 NMDG, 2.5 KCl, 1.2 NaH$_2$PO$_4$, 30 NaHCO$_3$, 20 N-2-hydroxyethylpiperaxine-N-2-ethanesulfonic acid (HEPES), 25 D-glucose, 2 Thiourea, 5 Na-ascorbate, 0.5 CaCl$_2$, and 10 MgSO$_4$ with a pH of 7.3–7.4. The HEPES ACSF contained (in mM): 92 NaCl, 2.5 KCl, 1.2 NaH$_2$PO$_4$, 30 NaHCO$_3$, 20 HEPES, 25 D-glucose, 2 Thiourea, 5 Na-ascorbate, 3 Na-pyruvate, 2 CaCl$_2$, and 2 MgSO$_4$. The recording ACSF contained (in mM): 129 NaCl, 3 KCl, 1.2 KH$_2$PO$_4$, 1.3 MgSO$_4$, 20 NaHCO$_3$, 2.4 CaCl$_2$, 3 HEPES, and 10 D-glucose.

**Sleep deprivation**. Wild-type mice in both genders were randomly assigned to three groups: mice in control group sleep freely without any interruption; mice with 6 h of sleep deprivation, which was started at ZT 6 (6 h after light on); mice with 12 h of sleep deprivation, which was started at ZT 0 (timepoint of light on). Sleep deprivation was performed via providing of new cages, novel toys and nesting materials, and gentle handling. All mice were sampled at ZT12 (timepoint of light off).

**Metabolomics profiling**. The controlling regions for sleep and wakefulness are located primarily in subcortical brain regions[13,15]. Therefore, tissues that contained the basal forebrain, thalamus, hypothalamus, midbrain, and brain stem for metabolomics profiling were collected from wild-type mice at ZT 5–7 (day) when mice were asleep (Sleep$_{day}$), and at ZT 17–19 (night) when mice were awake (Wake$_{night}$). Mice were deeply anesthetized with urethane and perfused with precooled PBS.

Tissues were dissected on ice and quickly frozen on dry ice and then transferred to –80 °C for storage. Metabolite profiling was performed by Metabolon, Inc. (Morrisville, NC, USA). Briefly, proteins were precipitated with methanol under vigorous shaking for 2 min (Glen Mills GenoGrinder 2000) followed by centrifugation. The sample extracts were placed briefly on a TurboVap (Zymark) to remove the organic solvent and stored overnight under nitrogen before preparation for analysis. Measurements were performed via a Waters ACQUITY ultra-performance liquid chromatography and a Thermo Scientific Q-Exactive high resolution/accurate mass spectrometer that was interfaced with a heated electro-spray ionization (HESI-II) source and Orbitrap mass analyzer operated at 35,000 mass resolution. Compounds were identified by comparison to library entries of purified standards or recurrent unknown entities. Peaks were quantified using area-under-the-curve analyses. Differential metabolites were defined when a $p$ value < 0.1 (by unpaired $t$-test) and a >1.3-fold change were observed between day and night values. Pathway analyses for these differential metabolites were performed using the web-based tool MetaboAnalyst 4.0 (https://www.metaboanalyst.ca). To generate the heatmap in Fig. 2b, metabolites belonging to the same species were summed together and z-scored between mice under different conditions.

**Lipidomics**. CX3CR1[CreER/+]:R26[iDTR/+] transgenic animals were administered tamoxifen and injected with DT 4 weeks later. Mice with the same genetic background that received DT injections, but not tamoxifen, served as controls. One day after the final DT injection, mice were deeply anesthetized with urethane when the mice were either awake (during the night) or asleep (during the day) and perfused with precooled 20 mM Tris-HCL (pH 7.5). The subcortical brain regions listed above were dissected on ice, quickly frozen on dry ice and then transferred to –80 °C for storage. Sleep-deprived mice and controls were sampled at ZT12 with the method as described above. For naïve mice with video recording before sampling, half of the subcortical region was used for ceramide measurement. Ceramide measurements were conducted by the Lipidomics Shared Resource Core at Stony Brook University. Briefly, brain samples were homogenized in 20 mM Tris-HCL (pH 7.5) with 1% protease/phosphatase inhibitors and centrifuged. The resulting supernatant was divided into two parts for lipidomic profiling and protein level measurements, respectively. Ceramide measurements were conducted by the Lipidomics Shared Resource Core at Stony Brook University. Briefly, extraction mix A (70% isopropanol:ethyl acetate = 2:3) and internal standard was added to the sample for lipid extraction. After vortexing and centrifugation, the supernatant extracts were dried with a nitrogen dryer and resuspended with mobile phase B solution (MeOH, 1 mM ammonium formate, 0.2% formic acid). The samples were measured via HPLC coupled to mass spectrometry (LC/MS)[49]. Protein concentrations were measured with BCA protein determination kits (Thermo scientific, 23225) according to the manufacturer's instructions. The amount of ceramide was normalized to the protein level.

**Histology**. Mice were deeply anesthetized with urethane and transcardially perfused with PBS followed by 4% paraformaldehyde. Brains were removed and postfixed in the same fixative at 4 °C overnight, then cryoprotected with 30% sucrose at 4 °C for 1 day. Coronal brain sections (50-μm thickness) were collected with a microtome. The sections were permeabilized in 0.25% Triton X-100 in 0.1 M PBS (0.25% PBST) for 0.5 h, followed by blocking solution (0.1% PBST with 1% of donkey serum) for 1 h and then incubated in blocking solution with primary antibody at 4 °C. Samples exposed to primary antibodies were incubated overnight. The primary antibodies used in the present study included: rabbit anti-Iba1 (1:1,000, 019-19741, Wako), goat anti-GFP (1:1,000, 600-101-215, Rockland), chicken anti-GFAP (1:1000, ab-4674, Abcam), mouse anti-GAD67 (1:500, MAB5406, Millipore), rabbit anti-PV (1:1,000, ab-11427, Abcam), mouse anti-ceramide (1:50, MID-15B4, Enzo), and rabbit anti-NeuN (1:500, ab-177487, Abcam). After incubation, the sections were washed three times in 0.1% PBST (5 min/wash) and immersed in 0.1% PBST with the corresponding secondary antibody with a dilution of 1:1000 for 3 h at room temperature. The secondary antibodies used in the present study included: donkey anti-goat 488 (A-11055, ThermoFisher), donkey anti-mouse 594 (A-21203, ThermoFisher), donkey anti-mouse 647 (A32787, ThermoFisher), donkey anti-rabbit 488 (711-545-152, JacksonImmunoResearch), and donkey anti-rabbit 594 (711-585-152, JacksonImmunoResearch), donkey anti-chicken (703-585-155, JacksonImmunoResearch). Finally, the sections were rinsed three times in 0.1 M PBS (5 min/rinse) and mounted onto micro slides. Images were captured with a confocal microscope using z-stack mode with 2 μm steps (FV1000, Olympus). All parameter settings for image collection were kept consistent between images.

For the validation of virus expression and implant locations, whole-brain images were collected. To validate the expression of virus (DIO-KORD, DIO-hM3Dq), we compare the viral expression region with the mouse brain atlas of the Allen Brain institute, and performed a 3D-reconstruction of the TRN for each mouse with a MATLAB-based function BrainMesh and marked TRN subregions with infected cell bodies with color (Supplementary Fig. 8). For validation of the efficiency of microglial depletions, CX3CR1[CreERT2/+]:R26[iDTR/+] mice were perfused on different days following either DT or vehicle injections, brain and peripheral organs (kidney and spleen) were collected for further analysis. As the CX3CR1 promoter controls expression of transgenes that encode tamoxifen-inducible Cre recombinase Cre-ERT2 and the downstream IRES-EYFP elements in

this transgenic line[27,30], microglia were identified as cells that were positive for both Iba1 and GFP. Consistent with a previous study[27,28,30], we also noticed a complete overlap between Iba1- and GFP-positive cells across all examined brain regions. CX3CR1-expressing GFP-positive cells were also observed and counted in the kidney and spleen. In general, 4–6 images were collected from each examined region for one animal. For microglial morphology analyses, images containing the cell bodies of measured microglia were stacked together, and then NeuronJ (a plugin available in ImageJ, National Institutes of Health, Bethesda, MD, USA) was used to trace all the processes of each microglia. The total process length of each microglia was quantified and compared for the different treatments. For a comparison of differences in microglia morphologies between $Sleep_{day}$ and $Wake_{night}$, CX3CR1$^{CreERT2/+}$ mice were perfused at ZT 5–7 (day) and ZT 17–19 (night). Animal states were maintained (i.e., asleep during the day and awake at night) before anesthesia. For testing the correlation between ceramide level, TRN microglial morphology and sleep architecture, behavior of naïve mice was monitored with video recording for 1 h before sampling, then the brain was been divided into two sagittal part, with subcortical region of one part been provided for lipidomic assay, the other part was used for immunostaining of Iba1. At least 20 Iba1-positive microglia were imaged from TRN region for each mouse, the mean value of total process length of examined microglia was used to analyze the correlation between subcortical ceramide level and animal behavior.

**Measurement of locomotor activity**. Four weeks after tamoxifen administration, CX3CR1$^{CreER/+}$:R26$^{iDTR/+}$ mice were exposed to gentle handling for at least 3 days (5 min/day) before the onset of experiments. Mice were randomly divided into two groups to receive either vehicle or DT injections. We placed a ceiling-mounted video camera above the recording area, and each mouse was introduced to this area and allowed to freely explore the area for 5 min of testing. Using an automated tracking system (Ethovision XT, Noldus), we were able to track the animal traces and count the number of times that each animal moved during each test period. The total distance traveled was calculated during the testing period for each animal. Locomotor activity was compared in the same mice before and after DT/vehicle injections.

**Reporting summary**. Further information on research design is available in the Nature Research Reporting Summary linked to this article.

## Data availability
The data that support the findings of this study are available in Source Data and Supplementary Tables. Other data are available upon reasonable request. Mouse brain atlas is available on Allen Institution (https://mouse.brain-map.org/experiment/thumbnails/100048576?image_type=atlas). Source data are provided with this paper.

## Code availability
The custom codes used for data analysis in this study are provided together with the source data.

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

## Acknowledgements
We would like to thank Dr. H Colognato, Dr. J Levine, and Ge and Xiong laboratory members for their valuable comments on the manuscript. We also thank Dr. C Mao for the gift of the Acer3$^{-/-}$ mice. This work was supported by the National Institutes of Health (DC016746 and DC017470 to Q.X.; NS089770, AG046875, and NS104868 to S.G.), and SUNY Stony Brook startup funding (to Q.X.).

## Author contributions
H.L., S.G., and Q.X. designed the experiments. H.L. performed most of the experiments and data analysis. X.W. performed the in vitro electrophysiological recordings. Lu C. performed part of the metabolomic analysis. Liang C. performed part of the tetrode recordings. H.L., S.T., S.G., and Q.X. wrote the manuscript.

## Competing interests
The authors declare no competing interests.
