## [Peer Review File · Nature Communications]

Reviewers' Comments:

Reviewer #1:

Remarks to the Author:

In this paper Liu et al, provide new evidence for the microglial involvement in the regulation of wakefulness and sleep. They employ an array of experimental approaches showing that microglia depletion increase transitions between wakefulness and NREM. Using a metabolomic analysis they revealed that ceramide influences microglia in the TRN. Combining chemogenetics and pharmacological manipulations they show that microglia can modulate stable wakefulness through anterior TRN neurons via ceramide signaling. Overall this is a very interesting and informative paper. There are a few issues that are of concern to me especially the controls.

1. From my understanding, controls do not receive DT. Is it possible that DT on its own, can account for some of the observed effects?
2. Similarly, it seems that the controls did not receive TMX.
3. For the chemogenetics, similarly, did the control receive CNO or SalB alone. For CNO its known that it can affect neuronal activity.
4. Another issue is the use of CX3CR1 as a marker for microglia. Although the authors mention some of the limitations that this marker. It is expressed also in the periphery and in the brain also on all infiltrating immune cells. Thus, the conclusion that the effects are based on microglia is limited. Thus, the caveat should be mentioned earlier in the text, and conclusions are limited accordingly. This does not diminish my excitement by the study, it just a significant limitation that needs to be acknowledged. It will also be helpful to show whether indeed any other immune populations are affected by their manipulation.

Minor:

- I. It would be helpful to clarify for the reader in the first paragraph of results (pg5 row 21) when addressing the transition between wake and NREM as (WN) and from NREM to wakefulness (NW) as significantly increased results, and no differences are shown in transition states between REM to wake (RW) or NREM to REM (NR).
- II. Please add a note why did you choose to focus on the TRN as other areas may be as relevant based on Fig 3A

Reviewer #2:

Remarks to the Author:

I have reviewed a previous version of the manuscript at another journal and below was my review at the time:

Reviewer #1:

Comments for the Author:

The paper by Liu et al. reports on a series of experiments involving microglia depletion, polysomnography, neural recordings and biochemistry to synthesize that microglia regulate wakefulness through the thalamic reticular nucleus (TRN). I find the study unconvincing and fundamentally-flawed. The data presented do not support the major conclusions and definitely do not justify the title.

Major:

1. The authors start off with a microglia depletion experiment followed by polysomnography to make the first link between microglia depletion and sleep. This is a fundamental flaw. Part of the reason why work from laboratories such as Yang Dan have been revolutionary for sleep neurobiology is temporal control. Meaning, one can make a tight link between neural activity patterns and arousal changes if 3 conditions are met: correlation (neural activity patterns change in a manner that correlates with arousal state), sufficiency: driving the circuit drives arousal one way, within a tight temporal window, and necessity: suppressing the circuit drives arousal in the opposite way, again, within an equivalent temporal window. None of these conditions are met in

this experiments and the reader is left wondering how many things have also changed from microglia depletion in addition to this one, potentially incidental finding related to arousal.

2. Going from the above to the TRN is a huge logical jump. Yes, the TRN is associated with sleep rhythm generation and perhaps to some aspects of arousal regulation but there are many other circuits that control arousal as well. Why focus on the TRN is unclear at that point in the manuscript. Also, the effect sizes of arousal and microglia, TRN and arousal are so small that one wonders whether these data are robust in the first place.

3. In Figure 3, it's completely unclear why the authors conclude that the TRN is the locus of microglia-induced arousal changes based on the fact that DREADDing the TRN reverses the phenotype. It's a simple gain of function that they are repeating from the previous figure, just on top of a different background.

4. Figure 4: same thing, these ceramide effects are unlikely to be TRN specific.

In the current version, the authors did a better job at presenting their findings in a more orderly manner. However, the critiques are unchanged from before, the findings in totality are not as clear as the authors try to demonstrate. The mechanistic evidence remains circumstantial. I would hope that the authors would go back and try to think a bit harder about the details of their work and do more substantial characterization of their findings. For example, a recent series of papers have shown that microglia express ATPases and may collaborate with astrocytes to elevate extracellular adenosine that suppresses neurons during sleep need accumulation. Could this be relevant to their work? I think expanding their list of potential mechanisms and ensuring the robustness of their main finding would go a long way to get this work ultimately successful (not just accepted at a journal, but also read by the community).

Reviewer #3:

Remarks to the Author:

The MS by Liu et al demonstrates the involvement of microglia in regulating arousal. While their homeostatic role has been documented the present MS is the very first to convincingly demonstrate their involvement in the control of brain states. The authors apply an interdisciplinary approach using state of the art techniques. The experiments performed are sound, the aesthetic and scientific quality of the figures is outstanding, the writing clear and the stats adequate. The MS presents a novel finding of outstanding importance that is of great interest to a broad audience.

Their main results are as follows:

1. Depleting microglia (global or focally in the thalamic reticular nucleus, TRN) reduces wakefulness
2. In normal mice the brain levels of ceramide decreased during wakefulness
3. Increasing brain ceramide concentrations decreased, decreasing their concentrations increased wakefulness
4. There are diurnal variations in microglial process length in some brain areas including the TRN
5. Application of ceramide leads to a retraction of microglia processes in TRN, similar results were found in *Acer3*^{-/-} mice known to have increased ceramide concentration
6. The activity of most TRN neurons is higher during the night where mice tend to be more in the awake state
7. Application of ceramide or depleting microglia leads to decreased TRN firing
8. Specific activation of anterior TRN neurons in microglia-depleted mice and inhibiting ceramide production restores wakefulness

While my overall opinion of the work is enthusiastic I have a number of observations/suggestions which might improve the MS.

1. In Fig3B the authors show a significant decrease in microglial processes only in the TRN, but in the fluorescent pic shown there seems to be richer signal in the treated mice TRN than the vehicle.

What is the explanation?

2. In my opinion the mouse sleep/W cycle is a lot less homogenous than the human therefore the day/night dichotomy is not the best way to quantify the sleep related correlation of metabolites, microglial processes and so on. Continuous EEG monitoring and sleep stage scoring would be a more appropriate measure. I am not saying the authors did not do this at some level i.e. quantified nr of bouts, S/W transitions and so on and I do realize there is more W during the night in these mice (Fig1C)

3. Similar logic applies to the diurnal variation of microglial processes. In addition, have the authors considered measuring ceramide concentrations and microglial processes in the TRN of sleep deprived mice?

4. The description of TRN physiology is too brief:

“Using tetrode recordings, we found that aTRN neurons displayed higher averaged firing rates at night compared to daytime firing rates in normal mice (Figure 3G). Whole-cell patch recordings in brain slices revealed that extracellular application of ceramide suppressed aTRN neuronal firing capacity (Figure 3H)”

The details of the recording and what is being compared could also be in some way part of the main text.

Minor issues:

P9 L6 mice instead of mouse

The authors could avoid using too much past tense in the discussion, present tense strengthens the points.

We thank the editor and reviewers for their careful reading and constructive remarks. We have performed additional experiments and extensively revised the manuscripts according to the critiques and suggestions. Please find below a detailed point-by-point response to all comments (reviewers' comments in grey and italic, our responses in black).

Reviewer #1 (Remarks to the Author):

In this paper Liu et al, provide new evidence for the microglial involvement in the regulation of wakefulness and sleep. They employ an array of experimental approaches showing that microglia depletion increase transitions between wakefulness and NREM. Using a metabolomic analysis they revealed that ceramide influences microglia in the

TRN. Combining chemogenetics and pharmacological manipulations they show that microglia can modulate stable wakefulness through anterior TRN neurons via ceramide signaling. Overall this is a very interesting and informative paper. There are a few issues that are of concern to me especially the controls.

We appreciate the reviewer's comments on our study. In response to the concerns, we have performed the suggested control experiments and analyses. The new data has been included in the revised manuscript. Details are elaborated in the following responses.

1. From my understanding, controls do not receive DT. Is it possible that DT on its own, can account for some of the observed effects?

We thank the reviewer for pointing out this important control. In our revised **Figure 1** and **supplementary Figure 2 & 3**, we have included one group of control mice with no DT receptor expression but received DT injections. In these control mice, we found no changes in wake/sleep structure that are observed in microglia-depleted mice. However, we agree with the importance of this piece of data.

In details, for assessing microglia depletion effects on wake and sleep, we included a group of control mice (CX3CR1^{Cre-ERT2/+}:R26^{iDTR/-}) that received the same TAM and DT injections (**Fig. 1a**). Because these mice do not have cre-inducible *iDTR* transgene, TAM treatment will not induce DT receptor expression and thus DT injection will not induce microglia depletion. As shown in **Figure 1 d-f** and **Supplementary Figure 2b** and **3a-c**, after DT injection these control mice displayed no change in total durations (**Fig. 1d**, **Supplementary Fig. 2b, 3a**), bout durations (**Fig. 1 e&f**), bout numbers (**Supplementary Fig. 3b**) of wake, NREM sleep (NREMs) and REM sleep (REMs), and no change in number of transitions between wake, NREMs and REMs (**Fig. 1g**).

Corresponding results are included on page 6 line 1-17.

2. Similarly, it seems that the controls did not receive TMX.

In response to this suggestion, in our revised **Figure. 1** and **supplementary Figure. 2 & 3**, we have included two groups of control mice with TAM treatment. In these control groups, we found neither TAM treatment nor DTR expression by themselves changed sleep architecture in the mice before onset of DT/vehicle injection.

One group of control mice is the same one described above in response to the question #1. The mice without *iDTR* transgene (CX3CR1^{Cre-ERT2/+}:R26^{iDTR/-}) have no DT receptor expression and received both TAM and DT injections.

Another group of control mice with *iDTR* transgene (CX3CR1^{Cre-ERT2/+}:R26^{iDTR/+}) received TAM and vehicle injection (**Fig. 1a**), thus in these mice DT receptor expression was turned on, but microglia were not depleted because of no DT injection. 4 weeks after TAM injection, the sleep architecture in mice with or without TAM

treatment (**Supplementary Fig. 2a**), and mice with or without DT receptor expression (**Supplementary Fig. 3d**) were comparable.

Corresponding results are included on page 6 line 1-19.

3. For the chemogenetics, similarly, did the control receive CNO or SalB alone. For CNO its known that it can affect neuronal activity.

We agree with the reviewer that it has been a concern of CNO/SalB having undesired effects. To exclude this possibility, as suggested by the reviewer, we have performed and included two groups of mice for CNO and SalB controls. The data are presented in the revised **Figure 5** and **Supplementary Fig. 7**. There were no significant changes in wake/sleep architecture between CNO/SalB and vehicle injections in mice without expression of the corresponding DREADD receptors.

In **Figure. 5e** and **Supplementary Figure 7 a&b**, we included a group of GAD2^{Cre} mice with expression of DIO-mCherry in aTRN and examined the change of wake/sleep architecture between vehicle injection and SalB injection. Compared to the vehicle injection, SalB injection in mice expressing only DIO-mCherry (but not KORD) had no effect on transition numbers between wake and NREMs (**Fig. 5e**), bout durations and bout numbers of wake and NREMs (**Supplementary Fig. 7 a&b**).

In **Figure. 5f** and **Supplementary Figure. 7 c&d**, we included a group of GAD2^{Cre} mice with expression of DIO-mCherry in aTRN and examined the change of wake/sleep architecture between vehicle injection and CNO injection. Previous study showed that CNO affects locomotor activity (1-10 mg/kg)^{1, 2} in mice without DREADD expression. To exclude the potential off-target effect of CNO, we chose the lowest CNO concentration (i.p., 0.3 mg/kg) that has been used in previous reports^{3, 4, 5, 6, 7, 8, 9, 10}. Compared to the vehicle injection, CNO injection in mice expressing only DIO-mCherry (but not hM3Dq) had no significant effect on transition numbers between wake and NREMs (**Fig. 5f**), bout durations and bout numbers of wake and NREMs (**Supplementary Fig. 7 c&d**).

Corresponding results are included on page 11 line 19 to page 12 line 20.

4. Another issue is the use of CX3CR1 as a marker for microglia. Although the authors mention some of the limitations that this marker. It is expressed also in the periphery and in the brain also on all infiltrating immune cells. Thus, the conclusion that the effects are based on microglia is limited. Thus, the caveat should be mentioned earlier in the text, and conclusions are limited accordingly. This does not diminish my excitement by the study, it just a significant limitation that needs to be acknowledged. It will also be helpful to show whether indeed any other immune populations are affected by their manipulation.

The reviewer has mentioned an important potential caveat on this transgenic line. Indeed, as mentioned by the reviewer, the CX3CR1 is also expressed in the peripheral and circulating immune cells, such as monocytes and myeloid¹¹. In response to this concern, we have performed additional experiments and included new analyses to examine the possibility that the observed effects in this study also include contributions of other CX3CR1-expressing cells. We have included the new data in the revised **Figure. 1** and **Supplementary Figure 1**. We also included a brief discussion regarding this concern.

We first used immunostaining to confirm that in our study condition, the brain resident CX3CR1-expressing cells are microglia. As presented in the **Figure 1c** and **Supplementary Figure 1c**, GFP (CX3CR1-expressing) cells in the brain are highly co-localized with Iba1 (microglia marker) staining, indicating that they are microglia, that is consistent with many previous studies^{12, 13, 14, 15, 16, 17, 18, 19}. We agree with the reviewer that

CX3CR1 is also expressed in circulating immune cells such as monocytes and macrophages¹¹. Under pathological conditions with compromised blood-brain barrier, such as in the brain of animal models of multiple sclerosis, there is an invasion of monocytes²⁰. However, under physiological conditions and together with the staining as shown in **Figure 1c**, the CX3CR1-expressing cells in the brain parenchyma are microglia.

Second, we perform analyses to determine potential involvement of peripheral CX3CR1-expressing cells. The peripheral CX3CR1-expressing immune cells have a short half-life (<1 week) and can completely renew the whole population within a short period (< 4 weeks)^{13,21}. Although there is a regional difference of microglia renewal speed, microglia in general have slower renewal rate than other immune cells^{13,14}. The shortest renewal speed of microglia was observed in the hippocampus and cerebellum, which takes at least 8 weeks¹⁷, and cortical microglia can survive across the whole lifespan of mouse^{17,22}. Therefore, we can take advantage of this difference in renewal speed and target microglia specifically. After TAM injection in CX3CR1^{Cre-ERT2/+}:R26^{iDTR/+} mice to induce DT receptor expression in CX3CR1-expressing cells, we waited for 4 weeks before DT injection following the strategy used in previous studies^{14,16,23}. During this 4 weeks, DTR-expressing microglia population will stay stable but almost all peripheral CX3CR1+/DTR+ immune cells will renew and should no longer express DT receptors. Thus, late onset of DT injection will only affect microglia but not other CX3CR1-expressing cells. To confirm this strategy, we quantify CX3CR1-expressing cells (GFP+) in brain, kidney, and spleen before and after DT injection. **Figure 1b** showed that after DT injection, brain GFP+ cells sharply dropped to ~27% of the baseline value, but there is no change in GFP+ cell numbers in kidney and spleen, indicating that DT induced depletion is preferentially targeting CX3CR1-expressing cells in the brain (i.e. microglia).

Third, we also examined potential involvement of other major factors in the brain following microglia depletion. Microglia rapidly repopulate within 1 week after the depletion, during which astrogliosis and cytokine storms were observed^{15,16,18}. In our study, we focused on the time point on which most microglia are depleted (one day after the final DT injection: Day 4, **Fig. 1c**), during which expression of TNF- α , interleukin-1 β and IL-6, and density of neuron and astrocyte has not been affected according to previous study^{15,16}. We confirmed that no change in densities of neuron and astrocyte after microglia depletion at this time point in our study (Day 4, **Supplementary Fig. 1 d-f**).

Altogether, we feel that the depletion strategy (inducible Cre line and 4-wk interval between cre induction and DT injection) allows us to target brain microglia preferentially, but we also discussed this potential limit in the text.

Corresponding results are included on page 4 line 21 to page 5 line 19, and discussion on page 15 line 1-11.

Minor:

I. It would be helpful to clarify for the reader in the first paragraph of results (pg5 row 21) when addressing the transition between wake and NREM as (WN) and from NREM to wakefulness (NW) as significantly increased results, and no differences are shown in transition states between REM to wake (RW) or NREM to REM (NR).

We thank the reviewer for this suggestion, we have included the following description in the main text (page 6 line 12-17)

“To further understand these vigilance dynamics, we analyzed the transitions between different brain states, the number of transitions from wake-to-NREMs (WN) and from NREMs-to-wake (NW) were significantly increased in microglia-depleted mice (**Fig. 1g, Supplementary Fig. 3b**). Interestingly, microglial depletion had no effect on REM sleep (**Fig. 1d-e, Supplementary Fig. 2d, 3a&b**) and number of transitions from NREMs-to-REMs (NR) and REMs-to-wake (RW) at night (**Fig. 1g**).”

II. Please add a note why did you choose to focus on the TRN as other areas may be as relevant based on Fig 3A

We have revised the text and clarify the rationale for choosing TRN as the focus (page 9 line 3 to page 10 line 5).

In brief, through microglia morphological analyses in brain regions involved in sleep/wakefulness regulation^{24, 25, 26}, we found five subcortical regions (MnPO, LPO, TRN, SCN and DM) in which microglia displayed diurnal morphological changes (**Fig. 4a**). Further experiment showed that among these five subcortical regions, only TRN microglia showed rapid response to external ceramide application (**Fig. 4b**). Further analyses showed that TRN microglia morphology exhibited a tight correlation with brain ceramide level (**Fig. 4c-e**). Therefore, we started to focus on TRN.

Reviewer #2 (Remarks to the Author):

I have reviewed a previous version of the manuscript at another journal and below was my review at the time:

Reviewer #1:

Comments for the Author:

The paper by Liu et al. reports on a series of experiments involving microglia depletion, polysomnography, neural recordings and biochemistry to synthesize that microglia regulate wakefulness through the thalamic reticular nucleus (TRN). I find the study unconvincing and fundamentally-flawed. The data presented do not support the major conclusions and definitely do not justify the title.

We appreciate the reviewer's effort in continuously reviewing our manuscript. We have now performed additional experiments and analyses based on the comments from all reviewers. The new data has been included. With new data, new analyses and revised logic flow to clarify the confusing points, we feel that the revised manuscript has been substantially improved. We have point-to-point responses below.

Major:

1. The authors start off with a microglia depletion experiment followed by polysomnography to make the first link between microglia depletion and sleep. This is a fundamental flaw. Part of the reason why work from laboratories such as Yang Dan have been revolutionary for sleep neurobiology is temporal control. Meaning, one can make a tight link between neural activity patterns and arousal changes if 3 conditions are met: correlation (neural activity patterns change in a manner that correlates with arousal state), sufficiency: driving the circuit drives arousal one way, within a tight temporal window, and necessity: suppressing the circuit drives arousal in the opposite way, again, within an equivalent temporal window. None of these conditions are met in this experiments and the reader is left wondering how many things have also changed from microglia depletion in addition to this one, potentially incidental finding related to arousal.

The reviewer has raised very interesting points. We appreciate the reviewer to share his/her opinions on the regulatory paths of sleep. We understand the reviewer's concerns locate on two aspects. One is on the temporal resolution used in this study, and the other is about the criteria for establishing the link between microglia and arousal changes. We addressed these two concerns separately below and tried to share our understandings on these concerns.

First, about the temporal resolution in the strategy used in our study. We totally agree with the reviewer that recently there are a cluster of elegant studies addressing how brain circuits promptly control sleep with the aid of available optogenetic and transgenic approaches. Dr. Yang Dan's group include many others have

significantly advanced the field. We start to have some pictures on how sleep/wake-related neuronal circuit induced sleep changes within a few seconds^{5, 6, 10, 27, 28}. In this study, we aimed at a slow pace of regulation. This study has been motivated by some daily experiences and previous studies. For example, it has been found that accumulation of somnogens takes a few hours to be effective on sleep^{29, 30}. In our daily experience, we know that Jetlag requires several days to alter the sleep-wake behavior³¹. This made us speculate that the mechanisms involved in regulating sleep expand in multiple systematic or temporal layers, from fine-tuned neural circuits controlling to circadian rhythm regulation, or from internal metabolic dynamics to external environment stimuli^{26, 29, 31, 32}. We feel that the reviewer will agree with us that our brain uses differential underlying mechanisms functioning in a wide-range timescale. Brain neurons may synchronize their activity with brain state transitions in milliseconds^{5, 10, 27}. On a slower end, microglia may survey the brain and sense the behavioral alterations in minutes^{12, 33}, and metabolites may react to these regulations from minutes to hours to regulate brain activities such as sleep^{34, 35}. Given the nature of this study on the slow pace of regulation, we tackled the microglia with a transgenic approach.

Additionally, we agree that the optogenetic manipulation provides important approaches for temporal controlling of neuronal activity. However, it was shown that laser stimulation itself produces strong off-target effects on microglial gene expression in mice lack of opsin expression³⁶. More importantly, optogenetic depolarization of microglial membranes showed a mild effect on morphology and surveillance function of microglia³⁷. Compared to optogenetics, chemogenetics act in a relatively slower time resolution, and affect cell activity mainly through influencing intracellular calcium or cAMP level^{38, 39}. However, microglia maintained low calcium activity and exhibited less correlation between cellular calcium transient and surveillance functions under steady state^{40, 41, 42, 43}. Therefore, conditional microglial depletion with high fidelity used in our study offers an efficient approach to reveal role of microglia in the brain function such as sleep. Microglia depletion has been successfully employed in previous studies for revealing roles of microglia on synaptic formation and pruning^{44, 45}, learning and forgetting^{14, 46}, social interaction and repetitive-behavior⁴⁷, chronic pain⁴⁸, and Alzheimer's disease and aging process^{49, 50}. We feel that this microglia depletion method is sufficient for addressing the question asked in this paper although a future-available method to control microglia with higher temporal resolution will be helpful for this study.

Second, about the criteria for establishing a link between microglia and arousal changes. We agree with the reviewer on the three key points to establish a link: correlation, sufficiency, and necessity. In the revised version, we have rewritten the text to clarify the evidence meeting these three criteria:

- (1) Correlation. In microglia-depleted mice: a) transitions between wake and NREMs increased (**Fig. 1d-g**), b) brain ceramide concentration increased (**Fig. 3a-b**), c) aTRN neuronal activity decreased (**Fig. 6b-f**). These results indicate a potential correlation among microglia depletion, wake-to-NREM transition, brain ceramide concentration, and aTRN neuronal activity.
- (2) Sufficiency. Elevation of brain ceramide in normal mice (without microglia depletion) increased wake-to-NREM transitions (**Fig. 3c**). Suppression of aTRN neuronal activity in normal mice increased wake-to-NREM transitions (**Fig. 5e**). It indicates both ceramide and aTRN neuronal activity is capable of modulating wake stability.
- (3) Necessity. In microglia-depleted mice, either suppression of ceramide accumulation (**Fig. 3d**) or activation of aTRN neuronal activity (**Fig. 6g-j**) block microglia-depletion effect on wake-to-NREM transitions.

Furthermore, we have included substantial evidence showing that microglia play a regulatory role on daily fluctuation of ceramide (**Fig. 3b**), which can suppress aTRN neuronal activity (**Fig. 4i**).

Taken together, we believe that our data strongly demonstrates that ceramide is at least one of the key signals in mediating microglia interaction with aTRN neuron, which modulates wake-to-NREM transitions.

2. Going from the above to the TRN is a huge logical jump. Yes, the TRN is associated with sleep rhythm generation and perhaps to some aspects of arousal regulation but there are many other circuits that control arousal as well. Why focus on the TRN is unclear at that point in the manuscript. Also, the effect sizes of arousal and microglia, TRN and arousal are so small that one wonders whether these data are robust in the first place.

After we received comments from previous submission to *Nature Neuroscience*, all the authors have discussed all the comments including the above one. Based on the comments, we have thought carefully of our conclusions from the data that we collected. We have revised extensively our conclusions before we submitted the manuscript to Nature Communications for consideration. However, we appreciated the comments from the previous set of reviewers.

In the current revision (Nature Communications), we have rearranged the logic flow and provided further evidence to justify the reason that we focused on TRN. In brief, we first presented our findings showing ceramide as a potential mediator for microglia-induced wake/sleep changes (**Fig. 2 & 3**). Next, through microglia morphological analyses in brain regions involved in sleep/wakefulness regulation^{24,25,26}, we found five subcortical regions (MnPO, LPO, TRN, SCN and DM) in which microglia displayed diurnal morphological changes (**Fig. 4a**). In further experiments, we found that among these five subcortical regions, only TRN microglia showed rapid response to external ceramide application (**Fig. 4b**). Our experiments also revealed that TRN microglia and ceramide were actively related under different conditions (**Fig 4c-e**). All together, these evidences motivated us to focus on TRN. Note that we have detailed these findings on page 9 line 3 to page 10 line 5.

Regarding the concern on the effect size of arousal and microglia, we presented the key analyses below:

- 1) Microglia depletion decreased wakefulness at night to 82.1 ± 5.5 % of baseline (relative to day 0 in same mouse; MG^{DTR+} DT $p = 0.013$; **Supplementary Fig. 2d**), whereas change in MG^{DTR-} DT control is 98.8 ± 3.4 % ($p = 0.68$; **Supplementary Fig. 2b**), and in MG^{DTR+} Veh. control is 103.4 ± 3.2 % ($p = 0.56$; **Supplementary Fig. 2c**).
- 2) To better compare the behavioral change between microglia-depleted mice and controls, we calculate the differences of wake state between day 0 and day 4 for each animal (day 4 – day 0). In microglia-depleted group (MG^{DTR+} DT), change is -10.9 ± 3.3 %; in MG^{DTR-} DT control group, change is -0.9 ± 1.9 %; in MG^{DTR+} Veh. control group, change is 2.1 ± 1.8 . We then compared the difference between these three groups using one-way ANOVA with Fisher's *post hoc* test. We found a significant difference between microglia-depleted group and control groups (vs MG^{DTR-} DT, $p = 0.025$; vs MG^{DTR+} Veh, $p = 0.005$).

Based on these statistics, we feel that the effective size of arousal and microglia can support the conclusion that microglial depletion reduced stable wakefulness at night.

Regarding the concern on the effect size of TRN and arousal, we listed the key analyses below:

- 1) We showed that SalB/KORD-mediated inhibition of aTRN neurons significantly increased wake-to-NREMs transitions by 159.1 ± 19.3 % (paired t-test, $p = 0.0024$), and NREMs-to-wake transition by 147.9 ± 19.2 % (paired t-test, $p = 0.0098$, **Fig. 5e**). There is no change in controls (wake-to-NREMs transition, 87.2 ± 7.9 %, paired t-test, $p = 0.22$; NREMs-to-wake transition, 87.7 ± 9 %, paired t-test, $p = 0.28$; **Fig. 5e**).
- 2) We showed that CNO/hM3Dq-mediated activation of aTRN neurons significantly decreased wake-to-NREMs transitions by 77.4 ± 5.7 % (paired t-test, $p = 0.0072$), and NREMs-to-wake transition by 72.6 ± 6 % (paired t-test, $p = 0.004$; **Fig. 5f**). There is no change in controls (wake-to-NREMs transition, 102.1 ± 7.8 %, paired t-test, $p = 0.86$; NREMs-to-wake transition, 99.1 ± 9.8 %, paired t-test, $p = 0.94$; **Fig. 5f**).

These statistics suggests that aTRN neurons modulated transitions between wake and NREMs.

3. In Figure 3, it's completely unclear why the authors conclude that the TRN is the locus of microglia-induced arousal changes based on the fact that DREADDing the TRN reverses the phenotype. It's a simple gain of function that they are repeating from the previous figure, just on top of a different background.

In the current version (with Nature Communications), we have performed new experiments and included new evidence and rearranged the figures to demonstrate that aTRN is the locus of microglia-induced arousal changes (with evidence meeting the three criteria suggested by the reviewer).

In brief, we **first** showed in **Figure. 4** and **Supplementary Figure. 5a**, TRN microglia specifically sensitive to manipulation of ceramide (**Fig. 4b&c**), and local depletion of microglia in the aTRN reproduced the arousal effects observed in mice with global microglia depletion (**Fig. 4f, Supplementary Fig. 5b**). We also showed that aTRN neuronal activity decreased sharply during wake-to-NREM transitions (**Fig. 5 a-d**). We further showed that microglia depletion caused suppression of aTRN neuronal activity (**Fig. 4h, Fig. 6 a&b**). These results established the correlation between aTRN neuron activity and microglia-induced arousal changes. **Second**, we showed that chemogenetic inhibition of aTRN neuronal activity in control mice mimics the local microglia depletion effect on arousal and activating aTRN neuronal activity caused opposite effects (**Fig. 5 e&f**). These results provide the sufficiency evidence for the aTRN neuron functions in regulation of wake-to-NREM transition. **Third**, chemogenetically activating aTRN neuronal activity in microglia-depleted mice blocked microglia depletion effect on arousal (rescue experiment). This result indicates that aTRN neuron is necessary to mediate microglia-induced arousal change. Taken together, we conclude that aTRN is the locus (or at least one of the key locus) of microglia-induced arousal changes.

4. Figure 4: same thing, these ceramide effects are unlikely to be TRN specific.

We agree with the reviewer that ceramide mediated microglia-neuron interaction is likely not the TRN only. However, in the response to question # 3 above, we have shown that TRN is the key locus of microglia-induced arousal changes. In these experiments as listed below, we showed that the ceramide plays an important/sufficient role in regulating microglia-induced arousal changes.

- 1) We first found that brain ceramide concentration not only displays diurnal variation (**Fig. 2a-d**), but also is highly correlated with the vigilance state of the brain (**Fig. 2 e&f**), with higher ceramide level concurrent with higher sleep demands.
- 2) We next showed that following microglia depletion which cause an increase in wake-to-NREM transitions, the diurnal variation of ceramide is abolished, and the ceramide concentration is increased at night (**Fig. 3a&b**). Elevation of brain ceramide concentration in normal mice mimics the microglia-depletion effect on arousal (**Fig. 3c**). Suppression of brain ceramide in microglia-depleted mice blocks the microglia depletion-induced arousal change (**Fig. 3d**).
- 3) Importantly, application of ceramide suppresses aTRN neuronal excitability (**Fig. 4h**), consistent with the finding that microglia depletion suppressed aTRN neuronal excitability (**Fig. 4i**) and activity (**Fig. 6b**).

Altogether, we showed that ceramide can mediate the interaction between microglia and neurons to regulate the microglia-induced arousal changes at least in TRN. However, we should point out that we agree with the reviewer that the ceramide may also affect other brain areas for other functions.

In the current version, the authors did a better job at presenting their findings in a more orderly manner. However, the critiques are unchanged from before, the findings in totality are not as clear as the authors try to demonstrate.

The mechanistic evidence remains circumstantial. I would hope that the authors would go back and try to think a bit harder about the details of their work and do more substantial characterization of their findings. For example, a recent series of papers have shown that microglia express ATPases and may collaborate with astrocytes to elevate extracellular adenosine that suppresses neurons during sleep need accumulation. Could this be relevant to their work? I think expanding their list of potential mechanisms and ensuring the robustness of their main finding would go a long way to get this work ultimately successful (not just accepted at a journal, but also read by the community).

We appreciate these great points. In the revised version of the manuscript, we have provided new data and rearranged the logic flow and figures to clarify our findings and conclusion.

Regarding the suggestion of mechanism underlying microglia-neuron interaction, our unbiased metabolomic screening indeed suggested adenosine represents another possible hit. However, we had decided to explore the ceramide due to the following reasons:

- 1) Adenosine, as a metabolite in purine metabolism, is one of the hits displaying significant diurnal variation identified from our metabolomics screening (**Supplementary Fig. 4c**). However, within the detected purine metabolites only 22% of them showed significant diurnal concentration differences (**a**). In contrast, 49% of detected sphingolipids that includes ceramide showed significant diurnal differences (**b**). Pathway analysis further confirmed that sleep/wake behavior shows a stronger effect on expression of sphingolipids compared to purines (**c**). Therefore, we decided to explore the sphingolipid pathway, which has never been extensively addressed in sleep.
- 2) Importantly, further lipidomic screening identified ceramide (**Fig. 3**) as the potential candidate mediating microglia-induced arousal change. Thus, in the following experiments (**Fig. 4 & 5**), we tested how ceramide may mediate microglia-neuron interaction and involve in arousal modulation.
- 3) Also, previous study showed that adenosine executes a strong suppression on excitatory synaptic transmission but not on inhibitory synaptic transmission⁵², which have discouraged us to examine the adenosine in TRN neurons since they are inhibitory neuron population. However, without further tests, we do not exclude the role of adenosine in mediating microglia-neuron interaction in TRN.

We appreciate the reviewer's concern. Our current study demonstrates that ceramide may mediate the microglia-neuron interaction that modulate arousal but does not exclude the possibility that there are other metabolites involved in this modulation. We have included a related discussion in the text (page 19 line 1-16): "Our study does not rule out the potential involvement of other signaling molecules, such as adenosine. Adenosine-related metabolites are essential modulators of microglial morphology and activity^{53, 54, 55}. A recent study reported that adenosine produced by microglia is crucial for the suppressive role of microglia on neuronal activity⁵⁶. Adenosine maintains a low level during NREMs and accumulates during wake and within REMs^{29, 34}. Manipulation of either brain adenosine level or its receptors, especially in the basal forebrain, produced a prominent effect on sleep/wake behavior^{29, 34, 57}. Thus, it is highly possible that adenosine may also involve the microglia modulation of stable wakefulness, by functioning in either parallel or joint pathways with ceramide signaling. Previous studies provided limited evidence about the relationship between ceramide and adenosine. Ceramide acts as a pro-inflammation factor⁵⁸, whereas adenosine is an anti-inflammation factor⁵⁹. Ceramide may affect

adenosine production by blocking mitochondrial ATP release⁶⁰. Adenosine suppresses tumor necrosis factor-induced nuclear factor- κ B (NF- κ B) activation but has less effect on ceramide-induced NF- κ B activation⁶¹. Adenosine executes a strong suppression on excitatory synaptic transmission, but not on inhibitory synaptic transmission⁵². Future studies may reveal whether and how adenosine (or other signaling molecules) participates in this microglia-mediated wakefulness modulation.”

Differential metabolisms between day and night. **a.** Day/night effect on purine metabolism. Pie chart indicated that 22% of purine metabolites shown a significant change between day and night. Volcano plot listed differential metabolites (>1.3 -fold change (day vs. night) and p -values < 0.10) in solid black circle. n.s., non-significant metabolites, they were indicated in open gray circle. **b.** Day/night effect on sphingolipid metabolism. Pie chart indicated that 49% of sphingolipid metabolites shown a significant change between day and night. Volcano plot listed differential metabolites in red circle (solid red circle, ceramides; open red circle, other sphingolipids). n.s., non-significant metabolites, they were indicated in open gray circle. **c.** Pathway analysis of differential metabolites between day and night. Day/night change significantly affected sphingolipid metabolism, but not on purine metabolism. Gray shade area indicated non-significant metabolism pathways between day and night.

Response to comments from reviewer #3 (Remarks to the Author):

The MS by Liu et al demonstrates the involvement of microglia in regulating arousal. While their homeostatic role has been documented the present MS is the very first to convincingly demonstrate their involvement in the control of brain states. The authors apply an interdisciplinary approach using state of the art techniques. The experiments performed are sound, the aesthetic and scientific quality of the figures is outstanding, the writing clear and the stats adequate. The MS presents a novel finding of outstanding importance that is of great interest to a broad audience.

Their main results are as follows:

- 1. Depleting microglia (global or focally in the thalamic reticular nucleus, TRN) reduces wakefulness*
- 2. In normal mice the brain levels of ceramide decreased during wakefulness*
- 3. Increasing brain ceramide concentrations decreased, decreasing their concentrations increased wakefulness*
- 4. There are diurnal variations in microglial process length in some brain areas including the TRN*
- 5. Application of ceramide leads to a retraction of microglia processes in TRN, similar results were found in *Acer3*^{-/-} mice known to have increased ceramide concentration*
- 6. The activity of most TRN neurons is higher during the night where mice tend to be more in the awake state*
- 7. Application of ceramide or depleting microglia leads to decreased TRN firing*
- 8. Specific activation of anterior TRN neurons in microglia-depleted mice and inhibiting ceramide production restores wakefulness*

While my overall opinion of the work is enthusiastic I have a number of observations/suggestions which might improve the MS.

We thank the reviewer for the appreciation of our work on the novelty and significance, and for the constructive comments. The point-to-point responses are listed below:

1. In Fig3B the authors show a significant decrease in microglial processes only in the TRN, but in the fluorescent pic shown there seems to be richer signal in the treated mice TRN than the vehicle. What is the explanation?

We thank the reviewer for pointing out this confusion. In the revised text, we have clarified the difference between microglia processes and Iba1 signal intensity in reflecting microglia activity (page 9 line 4-21).

In brief, Iba1 is a classic microglia marker which labels both resting and activated microglia⁶². Activated microglia display enhanced Iba1 fluorescence intensity and retracted processes^{62, 63, 64}. The activated microglia with retracted processes have larger cell bodies and thicker processes^{51, 64, 65, 66}, that also increase the visual effect of the Iba1 signals. Therefore, in exogenous ceramide-treated mice and *Acer3*^{-/-} transgenic mice, the dramatically increased Iba1 signals in TRN is consistent with the decreased microglia processes in TRN (**Fig. 4b & c, Supplementary Fig. 5a**). Both support the finding that TRN microglia activity is sensitive to ceramide concentration.

2. In my opinion the mouse sleep/W cycle is a lot less homogenous than the human therefore the day/night dichotomy is not the best way to quantify the sleep related correlation of metabolites, microglial processes and so on. Continuous EEG monitoring and sleep stage scoring would be a more appropriate measure. I am not saying the authors did not do this at some level i.e. quantified nr of bouts, S/W transitions and so on and I do realize there is more W during the night in these mice (Fig1C)

We agree with the reviewer that it would be ideal to quantify the sleep correlation of metabolites and microglia processes directly with wake/sleep instead of day/night. However, it has been a technical challenge for us to monitor *in vivo* metabolite concentrations and/or microglia processes during EEG recordings to identify

wake/sleep epochs. Currently, ceramide sensor has poor specificity and limited availability⁶⁷. Two-photon imaging that can monitor the microglia processes required head-fixed configuration of animals, which increases the complexity for data interpretation. We have tried single-photon imaging, however, the signal was not good enough to identify the thin processes of microglia.

Therefore, we tried to address this concern with an alternative strategy. **First**, the brain samples we collected for metabolite and microglia analyses (**Fig. 2-4**) were 1) all from sleeping mice at day or awake mice at night and 2) at time points mice having averaged 65% sleeping time or 25% sleeping time, respectively (**Supplementary Fig. 4d**). Thus, although mouse sleep/wake cycle is not exactly synchronized with day/night, the samples were collected from mice with big differences in sleep and wake periods. **Second**, we monitored the mouse behaviors through infra-red video recording for 1 hour before sampling. We found the percentage of wake periods in the last 1 hour before sample collection is highly correlated with the brain ceramide concentration (**Fig. 2f**), and the brain ceramide concentration is highly correlated with the length of TRN microglia processes (**Fig. 4d**). Altogether, we believe our analyses reveal the sleep/wake related correlations in metabolite concentration and microglia processes.

Corresponding results are included on page 7 line 20-22 and page 9 line 21 to page 10 line 2. We also included a discussion about the limit in the analyses in this study (page 15 line 15-18).

3. Similar logic applies to the diurnal variation of microglial processes. In addition, have the authors considered measuring ceramide concentrations and microglial processes in the TRN of sleep deprived mice?

We agree with the reviewer for the limitation of correlation analyses between sleep/wake behavior and microglial processes.

Beyond the response listed above (question #2), to further reveal the correlation between sleep/wake and ceramide concentration/microglia processes, in the revised manuscript, we included new results from sleep-deprived mice. We conduct 6 hours and 12 hours sleep deprivation in mice by continuous providing novel toys, nesting materials, new cage and gently handling. Control mice did not receive any perturbation before sampling. We sacrificed sleep-deprived and control mice at the timepoint of light off (ZT12) and collected brain samples for analyses. We found sleep deprivation has a gradient effect on both ceramide concentration (**Fig. 2e**) and TRN microglia morphology (**Fig. 4e**). For both ceramide concentration and TRN microglia processes, 6 hours sleep deprivation induced a trend of changes, while 12 hours sleep deprivation induced significant increase of ceramide concentration and decrease of microglia processes.

According to the classic two-process model of sleep, prolonged wakefulness continuously increases sleep demand (drive) of an individual with a homeostatic regulation^{32, 68, 69}. The shorter microglia processes and higher ceramide level found in mice with a stronger sleep demand (sleep-deprived mice, **Fig. 2e, Fig. 4e**), together with the correlation analysis (**Fig. 2f, Fig. 4d**) and ceramide measurements in microglia-depleted mice sampled at the night and wild-type mice sampled at the daytime (**Fig. 3a-b**), our results indicate that high sleep demand may cause high ceramide level, and high ceramide level will suppress TRN neuronal activity (**Fig. 4i**) which in turn increase transitions from wake to NREMs.

Corresponding results are included on page 7 line 14-20 and page 10 line 2-4.

4. The description of TRN physiology is too brief: "Using tetrode recordings, we found that aTRN neurons displayed higher averaged firing rates at night compared to daytime firing rates in normal mice (Figure 3G). Whole-cell patch recordings in brain slices revealed that

*extracellular application of ceramide suppressed aTRN neuronal firing capacity (Figure 3H)“
The details of the recording and what is being compared could also be in some way part of the main text.*

We thank the reviewer for pointing out this issue. We have revised the text accordingly at page 10 line 12-21, as shown below:

“.....we first recorded aTRN neuronal activity with tetrode recordings in mice during daytime and nighttime. We found that in average, aTRN neurons displayed higher firing rates at night compared to those at day in normal mice (**Fig. 4g**).”

“To test whether microglia depletion affects aTRN intrinsic excitability (such as firing capacity), we performed whole-cell patch recordings onto individual aTRN neurons in acute-prepared brain slices, by injecting various levels of currents and quantifying the elicited action potentials. We found that aTRN neuronal firing capacity was suppressed in microglia-depleted mice as compared to normal mice (**Fig. 4h**). In microglia-depleted mice, we have shown that nighttime ceramide concentration was relatively elevated (**Fig. 3b**). Consistently, extracellular application of ceramide suppressed aTRN neuronal firing capacity from normal mice (**Fig. 4i**).”

Minor issues:

P9 L6 mice instead of mouse

Corrected.

The authors could avoid using too much past tense in the discussion, present tense strengthens the points.

Corrected.

Reference

1. Gomez JL, *et al.* Chemogenetics revealed: DREADD occupancy and activation via converted clozapine. *Science* **357**, 503-507 (2017).
2. Mahler SV, Aston-Jones G. CNO Evil? Considerations for the Use of DREADDs in Behavioral Neuroscience. *Neuropsychopharmacology* **43**, 934-936 (2018).
3. Anacleit C, *et al.* The GABAergic parafacial zone is a medullary slow wave sleep-promoting center. *Nat Neurosci* **17**, 1217-1224 (2014).
4. Hayashi Y, *et al.* Cells of a common developmental origin regulate REM/non-REM sleep and wakefulness in mice. *Science* **350**, 957-961 (2015).
5. Weber F, Chung S, Beier KT, Xu M, Luo L, Dan Y. Control of REM sleep by ventral medulla GABAergic neurons. *Nature* **526**, 435-438 (2015).
6. Eban-Rothschild A, Rothschild G, Giardino WJ, Jones JR, de Lecea L. VTA dopaminergic neurons regulate ethologically relevant sleep-wake behaviors. *Nat Neurosci* **advance online publication**, (2016).
7. Harding EC, *et al.* A Neuronal Hub Binding Sleep Initiation and Body Cooling in Response to a Warm External Stimulus. *Current Biology* **28**, 2263-2273.e2264 (2018).
8. Ren S, *et al.* The paraventricular thalamus is a critical thalamic area for wakefulness. *Science* **362**, 429 (2018).

9. Yu X, *et al.* GABA and glutamate neurons in the VTA regulate sleep and wakefulness. *Nature Neuroscience* **22**, 106-119 (2019).
10. Zhang Z, *et al.* An Excitatory Circuit in the Perioculomotor Midbrain for Non-REM Sleep Control. *Cell*, (2019).
11. Lee M, Lee Y, Song J, Lee J, Chang S-Y. Tissue-specific Role of CX(3)CR1 Expressing Immune Cells and Their Relationships with Human Disease. *Immune Netw* **18**, e5-e5 (2018).
12. Nimmerjahn A, Kirchhoff F, Helmchen F. Resting Microglial Cells Are Highly Dynamic Surveillants of Brain Parenchyma in Vivo. *Science* **308**, 1314-1318 (2005).
13. Goldmann T, *et al.* A new type of microglia gene targeting shows TAK1 to be pivotal in CNS autoimmune inflammation. *Nat Neurosci* **16**, 1618-1626 (2013).
14. Parkhurst CN, *et al.* Microglia promote learning-dependent synapse formation through BDNF. *Cell* **155**, 1596-1609 (2013).
15. Elmore MRP, *et al.* CSF1 receptor signaling is necessary for microglia viability, which unmasks a cell that rapidly repopulates the microglia-depleted adult brain. *Neuron* **82**, 380-397 (2014).
16. Bruttger J, *et al.* Genetic Cell Ablation Reveals Clusters of Local Self-Renewing Microglia in the Mammalian Central Nervous System. *Immunity* **43**, 92-106 (2015).
17. Tay TL, *et al.* A new fate mapping system reveals context-dependent random or clonal expansion of microglia. *Nat Neurosci advance online publication*, (2017).
18. Huang Y, *et al.* Repopulated microglia are solely derived from the proliferation of residual microglia after acute depletion. *Nature Neuroscience*, (2018).
19. Cserép C, *et al.* Microglia monitor and protect neuronal function through specialized somatic purinergic junctions. *Science* **367**, 528-537 (2020).
20. Ransohoff RM. Microglia and monocytes: 'tis plain the twain meet in the brain. *Nature Neuroscience* **14**, 1098-1100 (2011).
21. Yona S, *et al.* Fate mapping reveals origins and dynamics of monocytes and tissue macrophages under homeostasis. *Immunity* **38**, 79-91 (2013).
22. Fuger P, *et al.* Microglia turnover with aging and in an Alzheimer's model via long-term in vivo single-cell imaging. *Nat Neurosci advance online publication*, (2017).
23. Appel JR, *et al.* Increased Microglial Activity, Impaired Adult Hippocampal Neurogenesis, and Depressive-like Behavior in Microglial VPS35-Depleted Mice. *J Neurosci* **38**, 5949-5968 (2018).
24. Weber F, Dan Y. Circuit-based interrogation of sleep control. *Nature* **538**, 51-59 (2016).
25. Scammell TE, Arrigoni E, Lipton JO. Neural Circuitry of Wakefulness and Sleep. *Neuron* **93**, 747-765 (2017).
26. Liu D, Dan Y. A Motor Theory of Sleep-Wake Control: Arousal-Action Circuit. *Annual Review of Neuroscience* **42**, 27-46 (2019).
27. Xu M, *et al.* Basal forebrain circuit for sleep-wake control. *Nat Neurosci* **18**, 1641-1647 (2015).

28. Gent TC, Bandarabadi M, Herrera CG, Adamantidis AR. Thalamic dual control of sleep and wakefulness. *Nature Neuroscience*, (2018).
29. Porkka-Heiskanen T, Strecker RE, Thakkar M, Bjørkum AA, Greene RW, McCarley RW. Adenosine: A Mediator of the Sleep-Inducing Effects of Prolonged Wakefulness. *Science* **276**, 1265-1268 (1997).
30. Porkka-Heiskanen T, Strecker RE, McCarley RW. Brain site-specificity of extracellular adenosine concentration changes during sleep deprivation and spontaneous sleep: an in vivo microdialysis study. *Neuroscience* **99**, 507-517 (2000).
31. Yamaguchi Y, *et al.* Mice Genetically Deficient in Vasopressin V1a and V1b Receptors Are Resistant to Jet Lag. *Science* **342**, 85-90 (2013).
32. Borbely AA. A two process model of sleep regulation. *Hum Neurobiol* **1**, 195-204 (1982).
33. Liu YU, *et al.* Neuronal network activity controls microglial process surveillance in awake mice via norepinephrine signaling. *Nature Neuroscience* **22**, 1771-1781 (2019).
34. Peng W, Wu Z, Song K, Zhang S, Li Y, Xu M. Regulation of sleep homeostasis mediator adenosine by basal forebrain glutamatergic neurons. *Science* **369**, eabb0556 (2020).
35. Keller M, *et al.* A circadian clock in macrophages controls inflammatory immune responses. *Proceedings of the National Academy of Sciences of the United States of America* **106**, 21407-21412 (2009).
36. Cheng KP, Kiernan EA, Eliceiri KW, Williams JC, Watters JJ. Blue Light Modulates Murine Microglial Gene Expression in the Absence of Optogenetic Protein Expression. *Sci Rep* **6**, 21172 (2016).
37. Laprell L, Schulze C, Brehme M-L, Oertner TG. Optogenetic control of microglia membrane potential reveals signal transduction in chemotaxis. *bioRxiv*, 2020.2005.2019.104109 (2020).
38. Yizhar O, Fenno Lief E, Davidson Thomas J, Mogri M, Deisseroth K. Optogenetics in Neural Systems. *Neuron* **71**, 9-34 (2011).
39. Roth BL. DREADDs for Neuroscientists. *Neuron* **89**, 683-694 (2016).
40. Eichhoff G, Brawek B, Garaschuk O. Microglial calcium signal acts as a rapid sensor of single neuron damage in vivo. *Biochim Biophys Acta* **1813**, 1014-1024 (2011).
41. Brawek B, Garaschuk O. Microglial calcium signaling in the adult, aged and diseased brain. *Cell Calcium* **53**, 159-169 (2013).
42. Tvrdik P, Kalani MYS. In Vivo Imaging of Microglial Calcium Signaling in Brain Inflammation and Injury. *Int J Mol Sci* **18**, (2017).
43. Umpierre AD, Bystrom LL, Ying Y, Liu YU, Worrell G, Wu L-J. Microglial calcium signaling is attuned to neuronal activity in awake mice. *eLife* **9**, e56502 (2020).
44. Paolicelli RC, *et al.* Synaptic Pruning by Microglia Is Necessary for Normal Brain Development. *Science* **333**, 1456-1458 (2011).
45. Miyamoto A, *et al.* Microglia contact induces synapse formation in developing somatosensory cortex. *Nature Communications* **7**, 12540 (2016).
46. Wang C, *et al.* Microglia mediate forgetting via complement-dependent synaptic elimination. *Science* **367**, 688-694 (2020).

47. Zhan Y, *et al.* Deficient neuron-microglia signaling results in impaired functional brain connectivity and social behavior. *Nat Neurosci* **17**, 400-406 (2014).
48. Peng J, *et al.* Microglia and monocytes synergistically promote the transition from acute to chronic pain after nerve injury. *Nat Commun* **7**, 12029 (2016).
49. Giorgetti E, *et al.* Modulation of Microglia by Voluntary Exercise or CSF1R Inhibition Prevents Age-Related Loss of Functional Motor Units. *Cell Rep* **29**, 1539-1554 e1537 (2019).
50. Spangenberg E, *et al.* Sustained microglial depletion with CSF1R inhibitor impairs parenchymal plaque development in an Alzheimer's disease model. *Nature Communications* **10**, 3758 (2019).
51. Prinz M, Jung S, Priller J. Microglia Biology: One Century of Evolving Concepts. *Cell* **179**, 292-311 (2019).
52. Qi G, van Aerde K, Abel T, Feldmeyer D. Adenosine Differentially Modulates Synaptic Transmission of Excitatory and Inhibitory Microcircuits in Layer 4 of Rat Barrel Cortex. *Cerebral cortex (New York, NY : 1991)* **27**, 4411-4422 (2017).
53. Davalos D, *et al.* ATP mediates rapid microglial response to local brain injury in vivo. *Nature Neuroscience* **8**, 752-758 (2005).
54. Haynes SE, *et al.* The P2Y₁₂ receptor regulates microglial activation by extracellular nucleotides. *Nat Neurosci* **9**, 1512-1519 (2006).
55. Orr AG, Orr AL, Li X-J, Gross RE, Traynelis SF. Adenosine A_{2A} receptor mediates microglial process retraction. *Nature Neuroscience* **12**, 872-878 (2009).
56. Badimon A, *et al.* Negative feedback control of neuronal activity by microglia. *Nature*, (2020).
57. Kumar S, Rai S, Hsieh K-C, McGinty D, Alam MN, Szymusiak R. Adenosine A_{2A} receptors regulate the activity of sleep regulatory GABAergic neurons in the preoptic hypothalamus. *American Journal of Physiology - Regulatory, Integrative and Comparative Physiology* **305**, R31-R41 (2013).
58. Maceyka M, Spiegel S. Sphingolipid metabolites in inflammatory disease. *Nature* **510**, 58-67 (2014).
59. Hasko G, Linden J, Cronstein B, Pacher P. Adenosine receptors: therapeutic aspects for inflammatory and immune diseases. *Nat Rev Drug Discov* **7**, 759-770 (2008).
60. Kong JN, *et al.* Novel function of ceramide for regulation of mitochondrial ATP release in astrocytes. *J Lipid Res* **59**, 488-506 (2018).
61. Majumdar S, Aggarwal BB. Adenosine suppresses activation of nuclear factor- κ B selectively induced by tumor necrosis factor in different cell types. *Oncogene* **22**, 1206-1218 (2003).
62. Ito D, Imai Y, Ohsawa K, Nakajima K, Fukuuchi Y, Kohsaka S. Microglia-specific localisation of a novel calcium binding protein, Iba1. *Molecular Brain Research* **57**, 1-9 (1998).
63. Sasaki Y, Ohsawa K, Kanazawa H, Kohsaka S, Imai Y. Iba1 Is an Actin-Cross-Linking Protein in Macrophages/Microglia. *Biochemical and Biophysical Research Communications* **286**, 292-297 (2001).
64. Saijo K, Glass CK. Microglial cell origin and phenotypes in health and disease. *Nat Rev Immunol* **11**, 775-787 (2011).
65. Hickman S, Izzy S, Sen P, Morsett L, El Khoury J. Microglia in neurodegeneration. *Nature Neuroscience* **21**, 1359-1369 (2018).

66. Salter Michael W, Beggs S. Sublime Microglia: Expanding Roles for the Guardians of the CNS. *Cell* **158**, 15-24 (2014).
67. Canals D, Salamone S, Hannun YA. Visualizing bioactive ceramides. *Chem Phys Lipids* **216**, 142-151 (2018).
68. Achermann P. The two-process model of sleep regulation revisited. *Aviat Space Environ Med* **75**, A37-43 (2004).
69. Borbely AA, Daan S, Wirz-Justice A, Deboer T. The two-process model of sleep regulation: a reappraisal. *J Sleep Res* **25**, 131-143 (2016).

Reviewers' Comments:

Reviewer #1:

Remarks to the Author:

The authors were able to address all the main concerns raised in my original review. The new data support previous findings and solidifies them.

Reviewer #2:

Remarks to the Author:

The manuscript is much better organized and the logical flow is clearer. That said, the manuscript would benefit from some editing to ensure that the language is clearer.

1. The authors refer to the night and day but these terms are irrelevant since mice are nocturnal, they should simply refer to either wakefulness and sleep or active and rest phases.

Two remaining conceptual issues:

1. Some mechanistic gaps are still there. It is unclear where ceramide comes from and whether it acts directly on microglia. Unclear whether microglia directly influence neural circuitry or other glia are involved.

2. The aTRN is not a homogenous entity; it contains several subnetworks that project to many Thalamic nuclei. I'm not sure to what extent these results can be reproducible without knowing those circuits with some more detail.

3. Related to the above— The authors should clarify how much the total state occupancy changes with the manipulations they have. Many things can cause state instability and I'm not sure why they report increase in both sleep to wake and wake to sleep transitions because I don't see how one can change without the other.

Reviewer #3:

Remarks to the Author:

This is a revised version of the MS where the authors have addressed all my concerns by performing difficult experiments, data analysis and revisions to the text. I am satisfied with the changes made which further strengthen my overall enthusiasm for the work.

We thank the reviewers for the comments and editor's decision on our revised manuscript. Point-to-point responses to the remaining comments are shown below.

Reviewer #2 (Remarks to the Author):

The manuscript is much better organized and the logical flow is clearer. That said, the manuscript would benefit from some editing to ensure that the language is clearer.

We appreciate the encouraging words. We have performed editing as suggested and have included several short additional paragraphs in the discussion to address these potential confusions scientifically. We also revisited and made sure of a clear set of stereotaxic coordinates in the methods so that other groups with similar interest could employ them directly. We expect with these suggested editing and additions the presentation has been greatly improved.

1. The authors refer to the night and day but these terms are irrelevant since mice are nocturnal, they should simply refer to either wakefulness and sleep or active and rest phases.

We do agree with the reviewer that the day/night association with sleep/wake in mice are opposite to human and other diurnal animals. In this respect, the change to 'either wakefulness and sleep or active and rest phases' will facilitate the reading. However, on another aspect, although mice spend most of time sleep at daytime and are more active at nighttime, there are sleep and wake epochs in both daytime and nighttime. To incorporate this suggestion and try to keep a precise reference, we revised the terms in Figures 2, 3, 4 and Supplementary Figure 4 as Sleep_{day} (sleep during daytime) and Wake_{night} (wakefulness during nighttime).

Two remaining conceptual issues:

1. Some mechanistic gaps are still there. It is unclear where ceramide comes from and whether it acts directly on microglia. Unclear whether microglia directly influence neural circuitry or other glia are involved.

We appreciate the reviewer to reemphasize these points and agree that they are important for us or other groups to address in the near future. We have included a paragraph in the discussion to review these mechanistic questions (page 19 lines 4-21). And attached below.

"In this study, we show that elevation of ceramide following microglial depletion decreased intrinsic excitabilities of aTRN neurons (**Figs. 4h & i, 6a-f**) and manipulations of ceramide productions either mimicked or blocked these microglia depletion effects (**Fig. 3c & d, 6g-j**), suggesting that microglia can modulate neuronal activity through ceramide. Ceramide locates in the branching point of sphingolipids metabolism¹, and most of brain cells may be able to produce ceramides. It remains unclear whether the ceramide critical for aTRN neuronal activity is produced from microglial cells or other cell types (e.g. astrocyte). Furthermore, whether and how ceramide directly interacts with neurons are largely unknown. Ceramide was reported to activate microglia^{2, 3} either through external application *in vitro* or stimulation within microglial cells. Sphingosine-1-phosphate (S1P) is a main catabolite of ceramide⁴. A recent work from our group revealed that sphingosine-1-phosphate receptor 2 (S1PR2) is specifically located in interneuron, with remarkable expression in interneuron enriched TRN region⁵. Manipulations of S1P signaling via S1PR2 tune inhibition level in the neural network⁵. Therefore, it is possible that microglia modulate aTRN neurons via ceramide-s1p-S1PR2 signaling pathway. Although we have shown that within the study time window the cell density of astrocytes was not changed (Supplementary Fig. 1d & e), it did not rule out the possibility that astrocytes and other brain cells are also involved in the modulatory process. The molecular and

cellular mechanisms underlying how microglia regulate subcortical ceramide concentrations, and how ceramide modulates neuronal excitability require further studies.”

2. The aTRN is not a homogenous entity; it contains several subnetworks that project to many Thalamic nuclei. I'm not sure to what extent these results can be reproducible without knowing those circuits with some more detail.

The reviewer has raised another very interesting point. The aTRN is indeed very heterogeneous with very sophisticated projections. It is definitely very interesting to find out which population and what circuit(s) play a major role in modulating the wakefulness stability as described in this study. To respond to this suggestion and more extensively elaborate the importance of this point, we first revisited and made sure of a clear set of stereotaxic coordinates (Page 24 line 3-9) in the methods so that other groups with similar interest could employ them directly. We also included a new paragraph to discuss our data and perspectives as pointed by the reviewer in our revised manuscript (page 17 lines 15-16, page 18 lines 9-17), and attached as below:

“TRN is a heterogeneous brain region with high molecular, cellular, anatomical, and functional diversity^{6,7}.”

“The aTRN neurons send projections to intermediodorsal, anterodorsal and dorsal midline thalamus^{8,9}. Those connections provide the structure basis for function of aTRN in selective attention, fear conditioning, and flight behavior^{9, 10, 11, 12}, for which stable wakefulness is a prerequisite. Based on the coordinates in this study, the recording and manipulations covered most of the aTRN (**Supplementary Figs. 6a, 8**). It is not clear whether all aTRN neurons or a subpopulation of aTRN neurons are essential for the phenotypes we observed. Furthermore, the aTRN mainly contains limbic and motor sector of TRN area^{6,7}, receiving projections from relevant thalamic nuclei (e.g. anterodorsal and laterodorsal thalamus) and cortical areas (e.g. prelimbic cortex and cingulate cortex)^{8,11}. The specific inputs and outputs of the aTRN that controlling wakefulness stability and the transition between wakefulness and NREM sleep, are of great interest for future studies.”

3. Related to the above— The authors should clarify how much the total state occupancy changes with the manipulations they have. Many things can cause state instability and I'm not sure why they report increase in both sleep to wake and wake to sleep transitions because I don't see how one can change without the other.

In response to the suggestion on the total state occupancy changes, we have re-visited the inclusion of the related analyses (**Figures 1d, 3d, 4f, 6i, and Supplementary Figures 2, 3a, 4e**).

In response to the question of “why they report increase in both...”: Increase in wake-to-NREMs transitions found in this study could be caused by either decreased wakefulness stability or facilitated sleep. If the manipulations in this study decreased wakefulness stability, and the NREMs maintenance remained intact (no change in NREMs bout duration), the NREM-to-wake transitions would be passively increased (this is what we have found in this study). However, if our manipulations facilitated NREMs and/or stabilized NREMs, the transitions from NREMs-to-wake were not necessarily increase. Therefore, we reported increase in both wake-to-NREM and NREM-to-wake, together with wake and NREM bout durations, to support our conclusions on wakefulness stability.

Referencies

1. Hannun YA, Obeid LM. Sphingolipids and their metabolism in physiology and disease. *Nature Reviews Molecular Cell Biology* **19**, 175 (2017).
2. Nakajima K, Tohyama Y, Kohsaka S, Kurihara T. Ceramide activates microglia to enhance the production/secretion of brain-derived neurotrophic factor (BDNF) without induction of deleterious factors in vitro. *Journal of Neurochemistry* **80**, 697-705 (2002).
3. Scheiblich H, Schlütter A, Golenbock DT, Latz E, Martinez-Martinez P, Heneka MT. Activation of the NLRP3 inflammasome in microglia: the role of ceramide. *Journal of Neurochemistry* **143**, 534-550 (2017).
4. Gault CR, Obeid LM, Hannun YA. An overview of sphingolipid metabolism: from synthesis to breakdown. *Adv Exp Med Biol* **688**, 1-23 (2010).
5. Wang X, *et al.* Metabolic tuning of inhibition regulates hippocampal neurogenesis in the adult brain. *Proceedings of the National Academy of Sciences* **117**, 25818-25829 (2020).
6. Vantomme G, Osorio-Forero A, Lüthi A, Fernandez LMJ. Regulation of Local Sleep by the Thalamic Reticular Nucleus. *Frontiers in Neuroscience* **13**, (2019).
7. Li Y, *et al.* Distinct subnetworks of the thalamic reticular nucleus. *Nature*, (2020).
8. Vantomme G, *et al.* A Thalamic Reticular Circuit for Head Direction Cell Tuning and Spatial Navigation. *Cell Rep* **31**, 107747 (2020).
9. Lee J-H, *et al.* The rostroventral part of the thalamic reticular nucleus modulates fear extinction. *Nature Communications* **10**, 4637 (2019).
10. Halassa MM, *et al.* State-dependent architecture of thalamic reticular subnetworks. *Cell* **158**, 808-821 (2014).
11. Dong P, *et al.* A novel cortico-intrathalamic circuit for flight behavior. *Nature Neuroscience* **22**, 941-949 (2019).
12. Nakajima M, Schmitt LI, Halassa MM. Prefrontal Cortex Regulates Sensory Filtering through a Basal Ganglia-to-Thalamus Pathway. *Neuron*, (2019).